# Apolipoprotein J is a hepatokine regulating muscle glucose metabolism and insulin sensitivity

Ji A Seo [1,2,17], Min-Cheol Kang [1,3,17], Won-Mo Yang [1,4,17], Won Min Hwang [1,5], Sang Soo Kim [1,6], Soo Hyun Hong[1,11], Jee-In Heo [2], Achana Vijyakumar[1], Leandro Pereira de Moura [1,12], Aykut Uner [1], Hu Huang [1,13], Seung Hwan Lee [1,14], Inês S. Lima[1,15], Kyong Soo Park [4], Min Seon Kim [7], Yossi Dagon[1], Thomas E. Willnow [8], Vanita Aroda[9,10,16], Theodore P. Ciaraldi [9,10], Robert R. Henry[9,10,18] & Young-Bum Kim [1✉]

Crosstalk between liver and skeletal muscle is vital for glucose homeostasis. Hepatokines, liver-derived proteins that play an important role in regulating muscle metabolism, are important to this communication. Here we identify apolipoprotein J (ApoJ) as a novel hepatokine targeting muscle glucose metabolism and insulin sensitivity through a low-density lipoprotein receptor-related protein-2 (LRP2)-dependent mechanism, coupled with the insulin receptor (IR) signaling cascade. In muscle, LRP2 is necessary for insulin-dependent IR internalization, an initial trigger for insulin signaling, that is crucial in regulating downstream signaling and glucose uptake. Of physiologic significance, deletion of hepatic ApoJ or muscle LRP2 causes insulin resistance and glucose intolerance. In patients with polycystic ovary syndrome and insulin resistance, pioglitazone-induced improvement of insulin action is associated with an increase in muscle ApoJ and LRP2 expression. Thus, the ApoJ-LRP2 axis is a novel endocrine circuit that is central to the maintenance of normal glucose homeostasis and insulin sensitivity.

[1] Division of Endocrinology, Diabetes, and Metabolism, Beth Israel Deaconess Medical Center and Harvard Medical School, Boston, MA, USA. [2] Division of Endocrinology, Department of Internal Medicine, Korea University College of Medicine, Seoul, Korea. [3] Research Group of Food Processing, Korea Food Research Institute, Wanju-gun, Jeollabuk-do, Korea. [4] Department of Molecular Medicine and Biopharmaceutical Sciences, Graduate School of Convergence Science and Technology, Seoul National University, Seoul, Korea. [5] Division of Nephrology, Department of Internal Medicine, College of Medicine, Konyang University, Daejeon, Korea. [6] Department of Internal Medicine and Biomedical Research Institute, Pusan National University Hospital, Busan, Korea. [7] Department of Internal Medicine, Asan Medical Center, University of Ulsan, College of Medicine, Seoul, Korea. [8] Molecular Cardiovascular Research, Max-Delbrueck-Center for Molecular Medicine, Berlin, Germany. [9] Veterans Affairs San Diego Healthcare System (9111 G), San Diego, CA 92161, USA. [10] Department of Medicine, University of California San Diego, La Jolla, CA 92093, USA. [11]Present address: Columbia University, New York, NY, USA. [12]Present address: School of Applied Science, University of Campinas, Limeira, Brazil. [13]Present address: East Carolina University, East Carolina Diabetes and Obesity Institute, Greenville, NC, USA. [14]Present address: College of Medicine, The Catholic University of Korea, Seoul, Korea. [15]Present address: Universidade Nova de Lisboa, Lisboa, Portugal. [16]Present address: Brigham and Women's Hospital and Harvard Medical School, Boston, MA, USA. [17]These authors contributed equally: Ji A Seo, Min-Cheol Kang, Won-Mo Yang. [18]Deceased: Robert R. Henry. ✉email: ykim2@bidmc.harvard.edu

Many metabolic organs mediate various aspects of inter-organ communication through secreted factors, including leptin[1], interleukin-6[2], FGF21[3], adiponectin[4], selenoprotein P[5], and ghrelin[6]. Among these molecules, liver-derived proteins, such as FGF21 and selenoprotein P, now known as hepatokines, can directly affect glucose and lipid metabolism in adipose tissue, liver, or muscle[7]. The identification of hepatokines has greatly advanced the field of metabolic physiology[3,7]. Over the past decade, researchers have put forth intensive efforts to identify critical mediators in the muscle that control glucose metabolism and insulin sensitivity[8,9]. However, it is unclear how peripheral tissue-derived circulating metabolic signals regulate insulin-mediated muscle metabolism.

Insulin resistance is a major risk factor for developing type 2 diabetes[10,11], and is associated with plasma lipid and lipoprotein abnormalities, including reduced HDL cholesterol and elevated triglyceride levels with increased hepatic secretion of triglyceride-rich VLDL[12]. In addition to these risk factors, various apolipoproteins particles, such as ApoCIII have gained increasing attention as new indicators for type 2 diabetes[13]. Apolipoprotein J (ApoJ, also called clusterin) was first described as a secreted sulfated glycoprotein that is ubiquitously found in metabolic tissues and body fluids[14]. ApoJ exists as multiple protein isoforms including the 75–80 kDa highly glycosylated secreted form, and the smaller intracellular forms that are not well characterized[15,16]. The secretory isoform of ApoJ is thought to have molecular chaperone activity depending on the degree of glycosylation[17,18] and binds to specific cell surface receptors in mediating its biological effects, such as endocytosis[19,20]. Although the exact role of ApoJ in many conditions remains unclear[15], ApoJ has been implicated in altered pathophysiologic disorders, including atherosclerosis[21], obesity[22], diabetes[23], and Alzheimer's disease[24].

ApoJ is found in plasma HDL and LDL, though its functions in lipoprotein metabolism are unclear, while nuclear-localized and cytoplasmic isoforms of ApoJ have been described, each with distinct functions[25]. Interestingly, a recent study reported that the serum ApoJ level closely correlates with insulin resistance and decreases according to improvement of insulin sensitivity in humans with type 2 diabetes[26], suggesting a functional link between ApoJ and insulin action. This is further supported by the findings that ApoJ in HDL is correlated with insulin sensitivity but ApoJ in LDL/VLDL is associated with insulin resistance[27]. However, a lack of knowledge of how ApoJ links with the metabolic actions of insulin in skeletal muscle has been a limitation in the field.

Low-density lipoprotein receptor-related protein-2 (LRP2, also called gp330/megalin) is a member of the LDL receptor family of lipoprotein receptors that bind to their extracellular ligands, such as ApoE and ApoJ before endocytotic uptake[19,20]. LRP2 also binds to leptin and mediates leptin transport crossing the choroid plexus[28]. In addition, LRP2 interacts with selenoprotein P and promotes its uptake in the proximal renal tubules of the kidney[29]. Although LRP2 has not been suspected to participate in obesity, a study with whole-exome sequencing analysis suggests that LRP2 could play a role in the development of early-onset obesity[30]. While important roles of LRP2 have been reported, nothing is known about the metabolic action of LRP2 on muscle glucose metabolism.

In this study, we investigate the physiological roles of ApoJ and LRP2 in insulin-dependent metabolic responses, with particular focus on sources and tissue targets of ApoJ in the context of interorgan crosstalk. Here, we identify the ApoJ → LRP2 axis coupled with the IR system as a key endocrine circuit regulating glucose homeostasis and insulin sensitivity.

## Results

**Liver is a major source of circulating ApoJ.** To determine tissue targets and/or sources of ApoJ, we generated mice lacking ApoJ in liver (liver-specific ApoJ-deficient mice (L-ApoJ$^{-/-}$)) by mating ApoJ-floxed mice with albumin-Cre transgenic mice. A lack of serum ApoJ was found in L-ApoJ$^{-/-}$ mice (Fig. 1a), suggesting

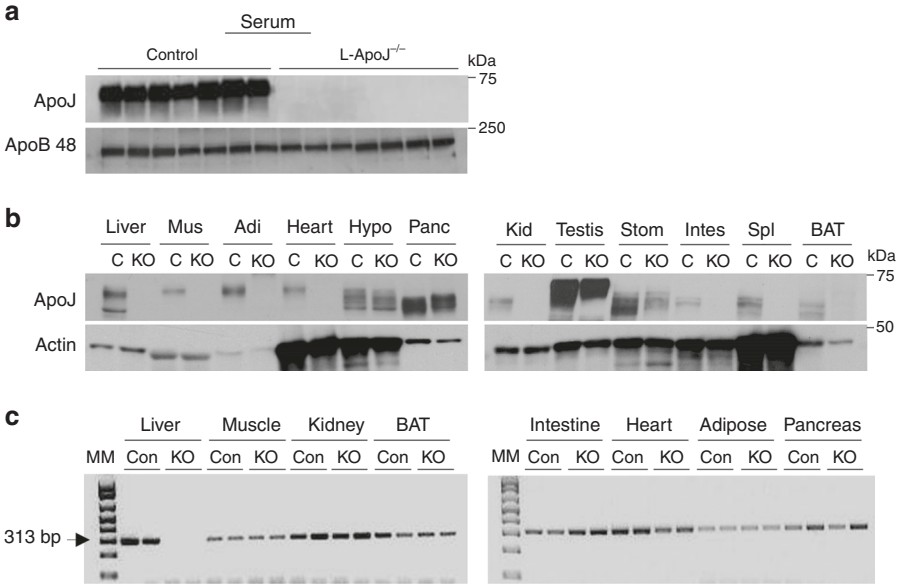

**Fig. 1 ApoJ levels in serum and multiple organs of L-ApoJ$^{-/-}$ mice. a** ApoJ protein levels in serum of liver-specific ApoJ-deficient mice (L-ApoJ$^{-/-}$). Serum was separated by SDS–PAGE. ApoJ or ApoB 48 was visualized by immunoblotting. These data are representative from more than three independent experiments. **b** ApoJ protein levels in multiple metabolic organs of L-ApoJ$^{-/-}$. Tissue lysates (20–50 μg) were separated by SDS–PAGE. ApoJ or actin was visualized by immunoblotting. These data are representative from more than three independent experiments. Mus muscle, Adi adipose tissue, Hypo hypothalamus, Panc pancreas, Kid kidney, Stom stomach, Intes intestine, Spl spleen, BAT brown adipose tissue; C: ApoJ$^{loxP/loxP}$ mice, KO: albumin-Cre: ApoJ$^{loxP/loxP}$ mice (L-ApoJ$^{-/-}$). **c** mRNA levels of ApoJ in multiple metabolic organs of L-ApoJ$^{-/-}$ mice. mRNA levels of ApoJ were measured by RT-PCR. Mice were studied at 12 weeks of age. Molecular marker (MM) is 100 bp DNA ladder. 313 bp indicates ApoJ mRNA. These data are representative from three independent experiments; Con: ApoJ$^{loxP/loxP}$ mice, KO: albumin-Cre: ApoJ$^{loxP/loxP}$ mice (L-ApoJ$^{-/-}$).

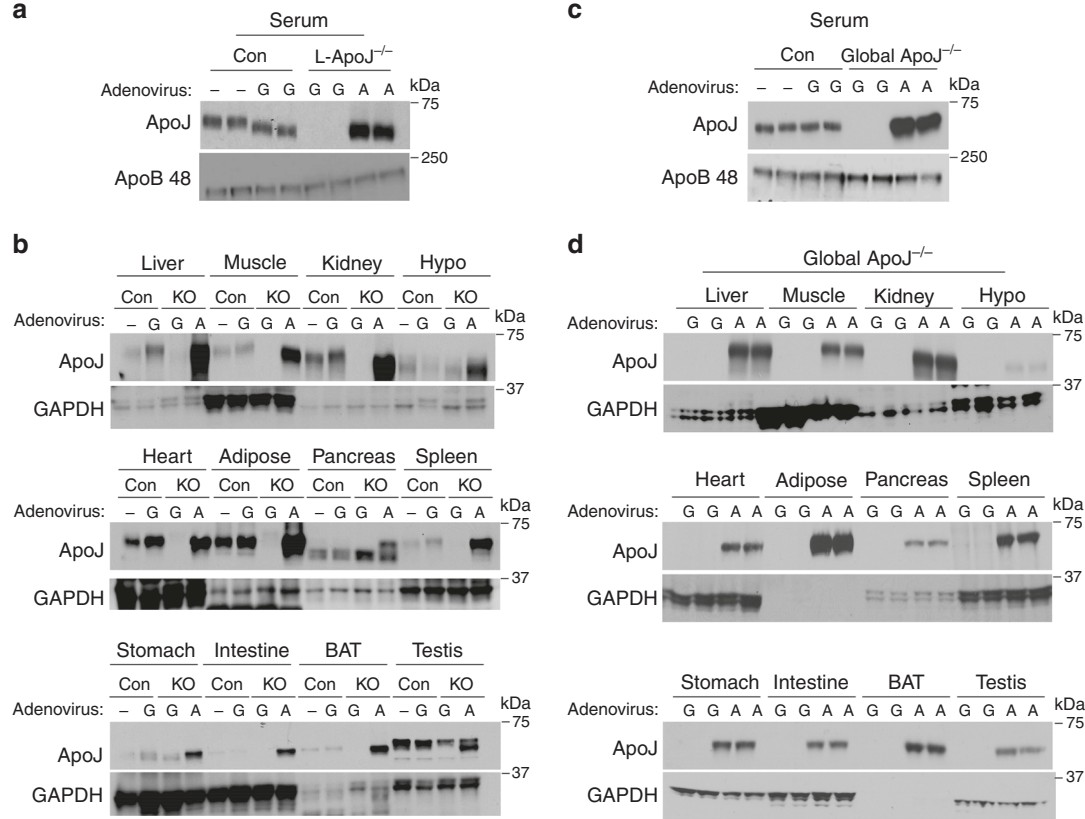

**Fig. 2 Liver is a major source of circulating ApoJ. a** ApoJ protein levels in serum of L-ApoJ$^{-/-}$ mice injected with ApoJ or GFP adenovirus. L-ApoJ$^{-/-}$ or global ApoJ$^{-/-}$ mice were injected with a recombinant adenovirus encoding a secretory ApoJ or GFP at a concentration of $2 \times 0^9$ pfu per gram of body weight via the tail vein. A: ApoJ-expressing adenovirus, G: GFP-expressing adenovirus. Serum was separated by SDS–PAGE. ApoJ, ApoB 48 or actin were visualized by immunoblotting. **b** ApoJ protein levels in multiple metabolic organs of in L-ApoJ$^{-/-}$ mice injected with ApoJ or GFP adenovirus. Tissue lysates (20–50 μg) were separated by SDS–PAGE. ApoJ or GAPDH were visualized by immunoblotting. **c** ApoJ protein levels in serum of in global ApoJ$^{-/-}$ mice injected with ApoJ or GFP adenovirus. **d** ApoJ protein levels in multiple metabolic organs of global ApoJ$^{-/-}$ mice. Tissue lysates (20–50 μg) were separated by SDS–PAGE. All data are representative from three independent experiments. Mice were studied at 9–10 weeks of age. Hypo hypothalamus, BAT brown adipose tissue.

that the liver is the major source of circulating ApoJ. In addition, levels of ApoJ were very low or absent in numerous metabolic organs of L-ApoJ$^{-/-}$ mice (Fig. 1b), which were not due to altered gene expression of ApoJ (Fig. 1c). Collectively, these data lead to the observation that ApoJ released by the liver into the circulation has access to multiple metabolic tissues, where ApoJ could play a role in the regulation of cellular metabolic events.

**Liver-derived ApoJ can transport into multiple metabolic organs.** To test the hypothesis that liver-derived ApoJ can be transported to metabolically active organs, an adenovirus carrying secretory ApoJ was administered to L-ApoJ$^{-/-}$ mice via the tail vein. Circulating ApoJ levels in L-ApoJ$^{-/-}$ mice injected with adenovirus ApoJ were greatly elevated compared with the mice injected with a control virus expressing green fluorescent protein (GFP; Fig. 2a). Consistently, increased ApoJ content was seen in multiple metabolic organs of L-ApoJ$^{-/-}$ mice injected with the adenovirus expressing ApoJ (Fig. 2b). These data were further confirmed with global ApoJ-deficient mice (ApoJ$^{-/-}$; Fig. 2c, d). Together, our results clearly demonstrate that liver-derived ApoJ can accumulate in metabolically active tissues, raising the possibility that hepatic ApoJ plays a key role in interorgan communication.

**Circulating ApoJ is normal in mice lacking ApoJ in muscle.** To further test the hypothesis that ApoJ protein in muscle is preserved in mice, where ApoJ was deleted in muscle, due to ApoJ

produced by the liver, muscle-specific ApoJ-deficient mice (M-ApoJ$^{-/-}$) were studied. We found that serum ApoJ levels did not differ between control and M-ApoJ$^{-/-}$ mice, indicating that muscle ApoJ is not a major source of circulating ApoJ (Fig. 3a). Importantly, M-ApoJ$^{-/-}$ mice had normal content of ApoJ protein in muscle and other metabolic organs (Fig. 3b). ApoJ mRNA was absent in muscle, whereas ApoJ mRNA was intact in the other organs (Fig. 3c). Our current results, combined with the results from L-ApoJ$^{-/-}$ mice injected with ApoJ adenovirus, suggest that ApoJ in muscle comes from circulating ApoJ produced by the liver, and the relative contribution of muscle ApoJ to total circulating ApoJ levels is quite limited. Rather, liver-derived ApoJ would represent the major source of ApoJ in muscle.

**L-ApoJ$^{-/-}$ mice display insulin resistance by impairing insulin signaling.** To determine the physiological function(s) of liver-derived ApoJ, we characterized the metabolic phenotypes of L-ApoJ$^{-/-}$ mice fed a normal chow diet. No significant differences emerged in body weight, fat mass, $O_2$ consumption, and $CO_2$ production between control and L-ApoJ$^{-/-}$ mice (Fig. 4a, Supplementary Fig. 2a–c). Blood glucose levels in the fasting state were significantly increased, but insulin levels were normal in L-ApoJ$^{-/-}$ mice compared with control mice (Fig. 4b, c). L-ApoJ$^{-/-}$ mice had normal circulating levels of IGF-1, IGFBP-3, and selenoprotein P (Figs. 4d, f, Supplementary Fig. 2f), but increased serum IGFBP-1 levels (Fig. 4e). Neither serum alanine

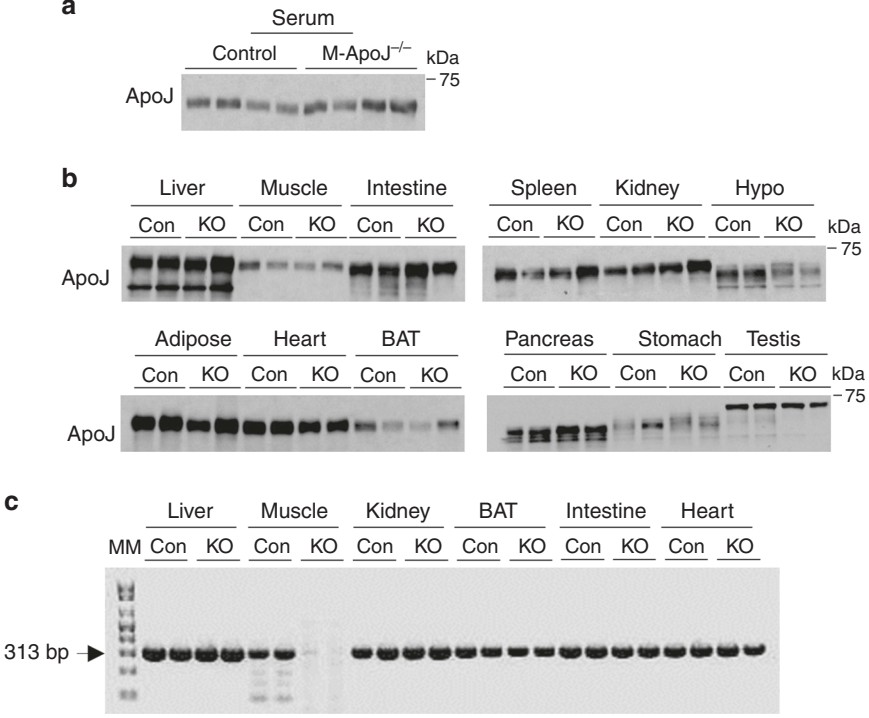

**Fig. 3 ApoJ levels are normal in serum and muscle of M-ApoJ$^{-/-}$ mice. a** ApoJ protein levels in serum of muscle-specific ApoJ-deficient mice (M-ApoJ$^{-/-}$). Serum was separated by SDS–PAGE. ApoJ was visualized by immunoblotting. **b** ApoJ protein levels in multiple metabolic organs of M-ApoJ$^{-/-}$. Tissue lysates (20–50 µg) were separated by SDS–PAGE. ApoJ was visualized by immunoblotting. **c** mRNA levels of ApoJ in multiple metabolic organs of M-ApoJ$^{-/-}$ mice. mRNA levels of ApoJ were measured by RT-PCR. Mice were studied at 8 weeks of age. Molecular marker (MM) is 100 bp DNA ladder. 313 bp indicates ApoJ mRNA. All data are representative from three independent experiments. Con: ApoJ$^{loxP/loxP}$ mice, KO: myogenin-Cre: ApoJ$^{loxP/loxP}$ mice (M-ApoJ$^{-/-}$).

aminotransferase (ALT), aspartate aminotransferase (AST), TG, nor cholesterol levels were altered in L-ApoJ$^{-/-}$ mice (Supplementary Fig. 2d, e, Fig. 4g–h). The contents of hepatic triglyceride and cholesterol were not different between control and L-ApoJ$^{-/-}$ mice (Fig. 4i–j). These data were confirmed by H&E staining of the liver (Fig. 4k). In addition, deletion of hepatic ApoJ had no effects on hepatic VLDL production and selenoprotein P levels, as well as hepatic gene expression of key enzymes involved in lipogenesis, gluconeogenesis, and glycolysis (Fig. 4l, Supplementary Fig. 2g–h). Interestingly, muscle LRP2 mRNA level was increased in L-ApoJ$^{-/-}$ mice compared with control mice (Supplementary Fig. 2i). This could be a compensatory effect, due to lack of its ligand.

Interestingly, selective deletion of hepatic ApoJ resulted in insulin resistance in the fasting state, as indicated by the fasting hyperglycemia, as well as failure to normalize glucose levels in the presence of elevated insulin levels. Such resistance was independent of adiposity (Fig. 4m). Of note, the slopes of the insulin tolerance test (ITT) curves are similar between control and L-ApoJ$^{-/-}$ mice (Fig. 4m), suggesting possible normal insulin sensitivity. However, L-ApoJ$^{-/-}$ mice still have reduced insulin responsiveness as maximal dose of insulin impairs insulin-stimulated glucose uptake and signaling. L-ApoJ$^{-/-}$ mice also displayed impaired glucose tolerance (Fig. 4n), which could result from impairment of glucose-stimulated insulin secretion (GSIS; Fig. 4o). We further determined the mechanism underlying hepatic ApoJ deficiency-induced insulin resistance by investigating insulin signaling in insulin-sensitive tissues. Interestingly, insulin-stimulated IR phosphorylation was markedly reduced by ~50–65% in muscle, adipose tissue, and liver in L-ApoJ$^{-/-}$ mice compared with control mice (Fig. 4p–r, Supplementary Fig. 2j–l). Subsequently, phosphorylation of downstream components of insulin signaling in skeletal muscle and adipose tissue, including

IRS-1/2, Akt, AS160, and GSK3, was also impaired in L-ApoJ$^{-/-}$ mice (Fig. 4p–r, Supplementary Fig. 2j–l). These data suggest that insulin resistance caused by hepatic ApoJ deficiency is most likely due to decreased insulin signaling in insulin-target tissues, primarily at the level of IR (Fig. 4p–r).

We further determined whether high-fat feeding could exacerbate the metabolic phenotype of L-ApoJ$^{-/-}$ mice. Similar to the results of normal chow diet, however, high-fat feeding had no effects on body weight, and body mass between control and L-ApoJ$^{-/-}$ mice (Supplementary Fig. 3a, b). Blood glucose levels were increased but serum insulin levels were normal in L-ApoJ$^{-/-}$ mice (Supplementary Fig. 3c, d). L-ApoJ$^{-/-}$ mice were insulin resistant and glucose intolerant, and displayed impaired insulin signaling (Supplementary Fig. 3e–i). Together, these findings demonstrate that hepatic ApoJ is involved in the regulation of whole-body glucose homeostasis and insulin sensitivity in vivo, and further suggest that a functional crosstalk exists between liver-derived ApoJ and insulin signaling in multiple tissues.

**LRP2 is a potential receptor of ApoJ in muscle.** Given that LRP2 is a potential receptor of ApoJ[19], it could be hypothesized that ApoJ levels would increase in serum and decrease in muscle if muscle LRP2 was deleted. We confirmed that LRP2 mRNA levels in muscle were significantly decreased (by ~80%) in muscle-specific LRP2-deficient mice (M-LRP2$^{-/-}$) compared with control mice, while LRP2 mRNA levels in other metabolic organs were normal (Supplementary Fig. 4b). The ability to detect muscle LRP2 protein by western analysis is constrained by the fact that its content is very low in muscle. However, using in situ proximity ligation assay (PLA), a powerful technology for detecting proteins with high specificity and sensitivity[31], we were able to observe LRP2 protein in muscle of control mice but not in

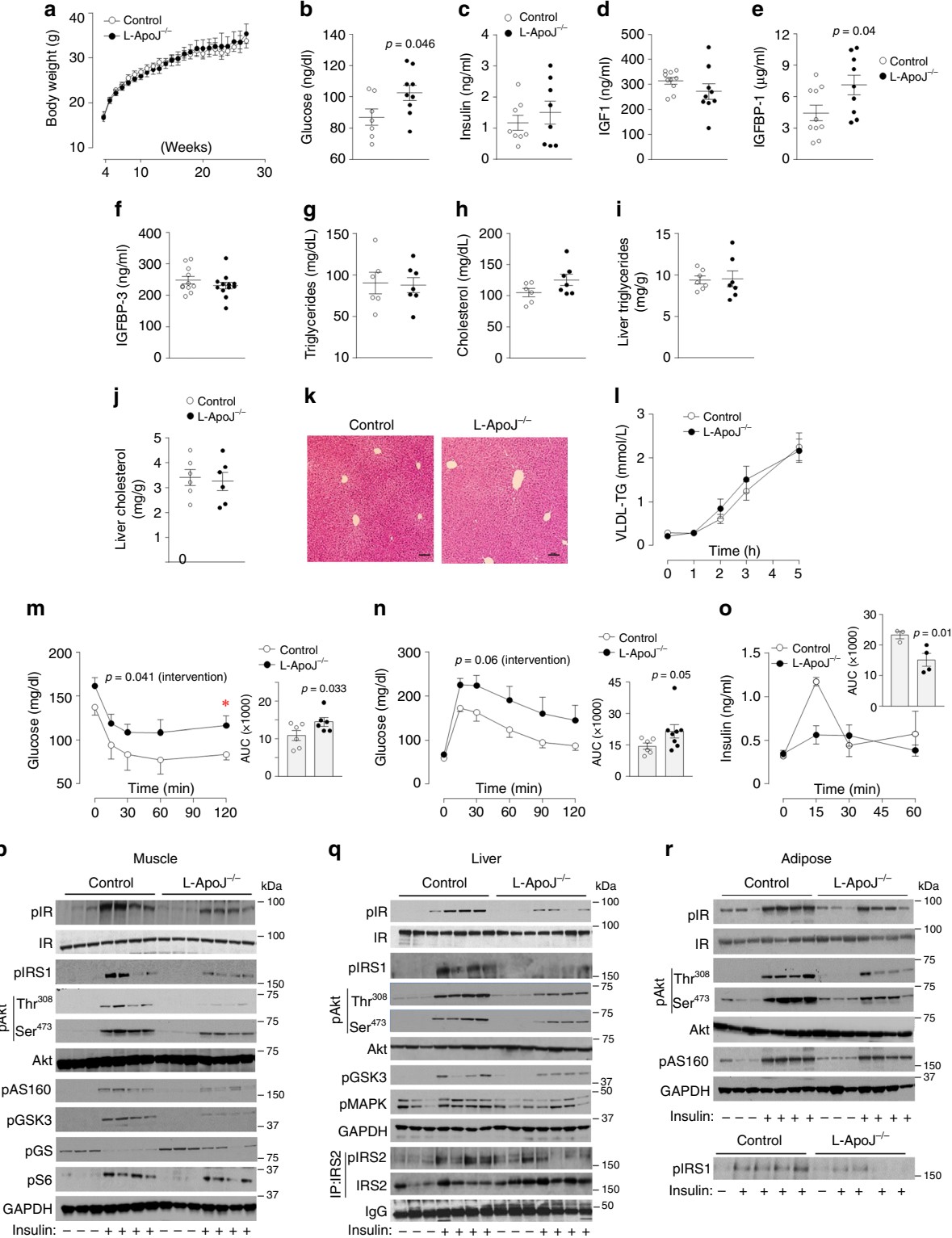

muscle of M-LRP2$^{-/-}$ mice (Supplementary Fig. 4e). In addition, endogenous ApoJ interacts with LRP2 in muscle of control mice: this effect was not detected in muscle of M-LRP2$^{-/-}$ mice (Supplementary Fig. 4f). These findings were confirmed by in vitro studies showing that exogenous ApoJ treatment significantly increased the interaction between ApoJ and LRP2 in $C_2C_{12}$ muscle cells, as evidenced by increases in the number of red spots at the cell surface of the cells. However, no interaction of LRP2

with insulin was found (Supplementary Fig. 4g), suggesting a specificity of ApoJ binding to LRP2.

Consistent with our hypothesis, serum ApoJ levels measured by ELISA were elevated by ~18% in mice lacking LRP2 in muscle (M-LRP2$^{-/-}$) compared to control mice (Fig. 5a). Concurrently, muscle ApoJ levels of M-LRP2$^{-/-}$ mice, determined by immunoblotting analysis, were lower than that of control mice (Fig. 5b). Normal levels of ApoJ were found in other metabolic

**Fig. 4 Loss of hepatic ApoJ leads to insulin resistance and glucose intolerance without obesity. a** Body weight ($n = 7$ for control, $n = 9$ for L-ApoJ$^{-/-}$), **b** fasting blood glucose ($n = 7$ for control, $n = 9$ for L-ApoJ$^{-/-}$), **c** fasting serum insulin ($n = 8$ per group), **d** serum IGF-1 ($n = 10$ for control, $n = 9$ for L-ApoJ$^{-/-}$), **e** serum IGFBP-1 ($n = 10$ for control, $n = 9$ for L-ApoJ$^{-/-}$), **f** plasma IGFBP-3 ($n = 11$ for control, $n = 12$ for L-ApoJ$^{-/-}$), **g** serum triglycerides ($n = 6$ for control, $n = 7$ for L-ApoJ$^{-/-}$), **h** serum cholesterol ($n = 6$ for control, $n = 7$ for L-ApoJ$^{-/-}$), **i** liver triglycerides ($n = 7$ per group), **j** liver cholesterol ($n = 6$ per group), **k** H&E-stained liver sections, **l** hepatic VDLD-TG production ($n = 5$ per group), **m** results of insulin tolerance test (ITT; $n = 6$ per group), **n** results of glucose tolerance test (GTT; $n = 6$ for control, $n = 8$ for L-ApoJ$^{-/-}$), **o** glucose-stimulated insulin secretion test (GSIS; $n = 3$ for control, $n = 4$ for L-ApoJ$^{-/-}$), **p** muscle insulin signaling, **q** liver insulin signaling, and **r** adipose tissue insulin signaling ($n = 7$ per group) were measured in control and L-ApoJ$^{-/-}$ mice. Serum parameters were measured from overnight fasted mice at 11–13 weeks of age. H&E-stained liver sections are representative from three independent experiments. The scale bars represent 100 µm. ITT was performed at 21 weeks of age and GTT was performed at 13 weeks of age. GSIS was performed at 16–18 weeks of age. Insulin signaling was performed at 25–26 weeks of age; AUC area under the curve. Logarithmic transformation was done for GTT. The AUC in GSIS was calculated by 30 min. All bars and errors represent means ± SEM. $p$ values by two-sided Student's $t$-test in **b** and **e**, and by repeated measures two-way ANOVA in **m** and **n** are indicated. $p$ values for AUCs were evaluated by one-sided Student's $t$-test for **m** and **n**, and by Mann–Whitney $U$ test for **o** based on the distribution of the data. $*p < 0.05$ vs control by repeated measures two-way ANOVA.

organs of M-LRP2$^{-/-}$ mice (Fig. 5c). Our results demonstrate that circulating ApoJ is retained in muscle via LRP2 and that LRP2 signaling could play a role in the maintenance of ApoJ homeostasis in circulation, at least in part.

**Muscle LRP2 deficiency leads to insulin resistance resulting from impaired insulin signaling.** To determine the role of LRP2 in muscle in control of glucose metabolism, we characterized the metabolic phenotypes of M-LRP2$^{-/-}$ mice fed a normal chow diet. Muscle-specific deletion of LRP2 had no effect on body weight or blood glucose levels (Fig. 5d, e). However, M-LRP2$^{-/-}$ mice had a higher risk for insulin resistance, as revealed by higher serum insulin levels in the fasting state (Fig. 5f). Correspondingly, M-LRP2$^{-/-}$ mice displayed insulin resistance (Fig. 5g), as evidenced by a blunted decrease in blood glucose levels after insulin injection over control mice. Glucose tolerance was also impaired in M-LRP2$^{-/-}$ mice compared with control mice (Fig. 5h). GSIS did not differ between control and M-LRP2$^{-/-}$ mice (Fig. 5i).

To explore the mechanism(s) by which deletion of muscle LRP2 contributes to systemic insulin resistance, we measured the ability of insulin to activate insulin signaling in insulin-sensitive tissues of M-LRP2$^{-/-}$ mice. Consistent with the results from L-ApoJ$^{-/-}$ mice, insulin-stimulated IR phosphorylation was decreased by ~50% in skeletal muscle of M-LRP2$^{-/-}$ mice, compared with control mice (Fig. 5j, Supplementary Fig. 4c). However, insulin-induced IR phosphorylation in the liver and adipose tissue of M-LRP2$^{-/-}$ mice was normal (Fig. 5j). In M-LRP2$^{-/-}$ mice, insulin's ability to increase IRS-1 and Akt phosphorylation were also markedly impaired in muscle, but unaltered in the liver and adipose tissue (Fig. 5j). These data suggest that LRP2 signaling is necessary for the regulation of insulin-mediated signaling in skeletal muscle and further provides in vivo evidence that ApoJ signaling is functionally linked with insulin signaling.

**Insulin-stimulated glucose uptake is impaired in L-ApoJ$^{-/-}$ and M-LRP2$^{-/-}$ mice.** To determine whether hepatic ApoJ deletion or muscle LRP2 deficiency leads to impaired insulin-mediated glucose uptake, we measured in vivo glucose uptake in muscle, liver, and adipose tissue.

Notably, insulin stimulation of glucose uptake was markedly decreased by ~30%, ~35%, and ~20% in muscle, adipose tissue, and brown adipose tissue in L-ApoJ$^{-/-}$ mice compared with control mice, respectively (Fig. 6a). Similarly, deletion of muscle LRP2 led to a ~23% reduction in insulin-induced glucose uptake in muscle, but not in adipose tissue or brown adipose tissue (Fig. 6b). Given that defective insulin-mediated glucose transport in skeletal muscle is a major contributor to the pathogenesis of insulin-resistant states[32], it is likely that systemic insulin resistance caused by hepatic ApoJ deficiency or muscle LRP2

deletion results primarily from decreased insulin-stimulated glucose uptake in skeletal muscle. Collectively, our findings highlight that hepatic ApoJ and muscle LRP2 are required for optimal insulin-mediated glucose uptake in muscle and further implicate ApoJ → LRP2 signaling is critical for maintaining normal glucose homeostasis.

**LRP2 is required for insulin-induced insulin receptor internalization.** To determine the mechanism(s) by which LRP2 is involved in the regulation of insulin-mediated glucose metabolism, we measured IR internalization, Akt phosphorylation, and glucose uptake in $C_2C_{12}$-myc-Glut4 cells, where LRP2 was knocked down. We confirmed that LRP2 mRNA levels were reduced by ~86% in LRP2 siRNA-transfected cells compared with control siRNA-transfected cells (Supplementary Fig. 4d). IR undergoes internalization upon ligand stimulation[33]. As expected, insulin rapidly stimulated IR internalization with ~35–50% reduction of the cell surface IR by 60 min (Fig. 7a); this effect was abolished by LRP2 inhibition (Fig. 7b). Consistent with the results in M-LRP2$^{-/-}$ mice, insulin-induced Akt phosphorylation, and glucose uptake were markedly decreased in $C_2C_{12}$-myc-Glut4 cells when LRP2 expression was suppressed (Fig. 7c, d). These data clearly demonstrate that LRP2 is required for insulin-induced IR internalization, which is crucial in regulating downstream signaling and glucose uptake.

Given that LRP2 is an endocytic receptor[19,20,22], we tested if clathrin-mediated endocytosis is involved in the regulation of insulin signaling. In response to insulin, FOXO1 nuclear export (from nucleus to cytosol), an indicator of activation of the PI3K/Akt signaling pathway[34], was markedly increased in $C_2C_{12}$ muscle cells (Fig. 7e). However, this response was completely blocked when muscle cells were treated with chlorpromazine or cytochalasin D, an inhibitor of clathrin-mediated endocytosis, (Fig. 7e), indicating that endocytosis is an essential step for this insulin-stimulated response.

**Pioglitazone reduces circulating ApoJ levels but increases muscle ApoJ and LRP2 in humans with insulin resistance.** An example of a relationship between elevated ApoJ levels and insulin resistance is seen in the polycystic ovary syndrome (PCOS). Women with PCOS display insulin resistance in both the fasting (HOMA2-IR) and insulin-stimulated glucose disposal rate (GDR) states compared to BMI-matched normal cycling control subjects (Table 1). Of note, women with PCOS have elevated (~35%) serum ApoJ levels over healthy controls (Fig. 8a). As expected, pioglitazone treatment of the PCOS subjects improved insulin action, evidenced by decreased HOMA2-IR and increased GDR (Fig. 8b, c, Table 1). Along with this, circulating ApoJ levels in these subjects were reduced and essentially normalized (Fig. 8d); placebo was without effect on either characteristic. Serum

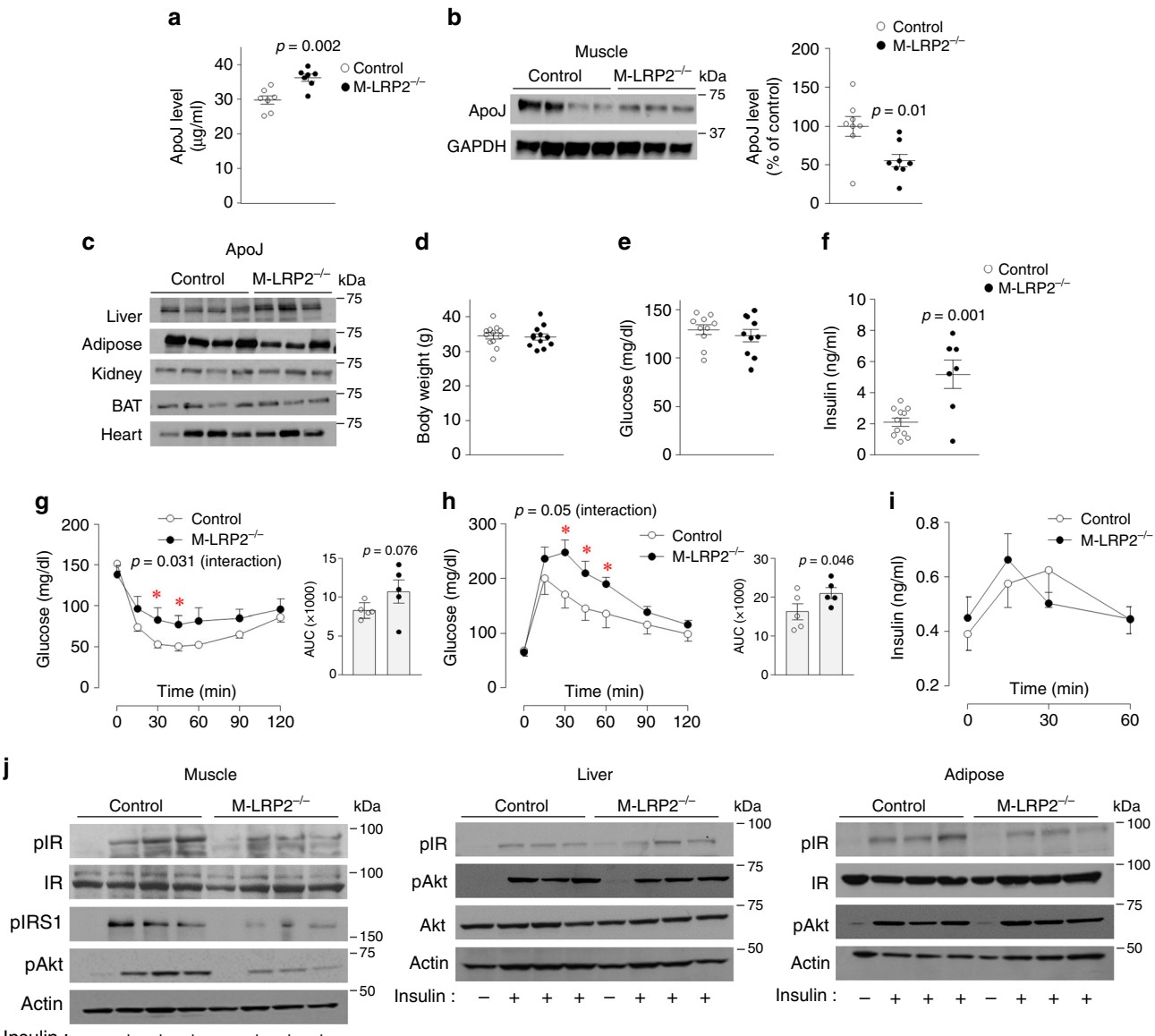

**Fig. 5 LRP2 deletion in muscle increases circulating ApoJ and causes insulin resistance and glucose intolerance. a** ApoJ protein levels in serum of muscle-specific LRP2-deficient mice (M-LRP2$^{-/-}$). Serum ApoJ levels were measured by ELISA. ApoJ, or ApoB 48 were visualized by immunoblotting. Scatter diagram shows densitometric quantitation of ApoJ protein from control and M-LRP2$^{-/-}$ mice ($n = 7$ per group). **b** ApoJ protein levels in muscle of M-LRP2$^{-/-}$. Muscle lysates (20 μg) were separated by SDS–PAGE. ApoJ or GAPDH was visualized by immunoblotting. Scatter diagram shows densitometric quantitation of ApoJ protein from control and M-LRP2$^{-/-}$ mice ($n = 8$ per group). **c** ApoJ protein levels in multiple metabolic organs of M-LRP2$^{-/-}$. Tissue lysates (20–50 μg) were separated by SDS–PAGE. ApoJ was visualized by immunoblotting. **d** Body weight ($n = 12$ per group), **e** random blood glucose ($n = 10$ per group), **f** serum insulin ($n = 11$ for control, $n = 7$ for M-LRP2$^{-/-}$), **g** insulin tolerance test (ITT; $n = 5$ per group), **h** glucose tolerance test (GTT; $n = 5$ per group), **i** glucose-stimulated insulin secretion test (GSIS; $n = 4$ per group), and **j** insulin signaling ($n = 4$ per group) were measured in control and M-LRP2$^{-/-}$ mice. Serum and muscle ApoJ level, body weight, and random blood glucose were measured at 16 weeks of age. Serum insulin levels were measured from overnight fasted mice at 24 weeks of age. ITT was performed at 24 weeks of age and GTT was performed at 18 weeks of age. GSIS was performed at 22 weeks of age. Insulin signaling were performed at 24 weeks of age. All bars and errors represent means ± SEM. $p$ values by two-sided Student's $t$-test in **a**, **b**, and **f**, and by repeated measures two-way ANOVA in **g** and **h** are indicated. $p$ values for AUCs were evaluated by one-sided Student's $t$-test for **g** and **h**. $p$ values for AUCs were evaluated by one-sided Student's $t$-test for **g** and **h**. $p$ values for interaction in **g** and **h** were obtained by Greenhouse–Geisser correction. *$p < 0.05$ vs control by repeated measures two-way ANOVA.

ApoJ levels in healthy and PCOS subjects were positively correlated with fasting insulin, free fatty acid, HOMA2-IR, and HOMA-β (%), and negatively correlated with GDR (Fig. 8e). However, there was no association between serum ApoJ levels and age, BMI, HDL cholesterol, LDL cholesterol, triglycerides, and adiponectin levels (Fig. 8e, Supplementary Fig. 5). While skeletal muscle, ApoJ protein levels and LRP2 mRNA in PCOS subjects at baseline were not different compared to normal

cycling subjects (Fig. 8f, g), pioglitazone treatment of PCOS subjects increased ApoJ protein content, as well as mRNA for LRP2 (Fig. 8h, i).

## Discussion

A major challenge in the field of metabolic physiology has been to understand the interorgan communication networks linked to

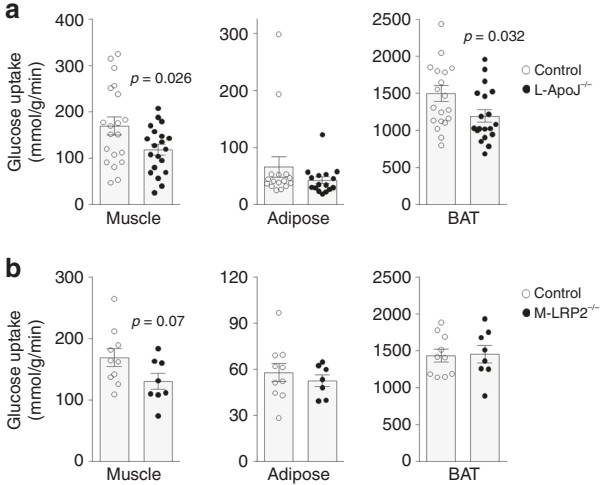

**Fig. 6 ApoJ → LRP2 signaling is required for insulin-stimulated glucose uptake. a** Insulin-stimulated glucose uptake in muscle, adipose tissue, and brown adipose tissue (BAT) was measured in control and L-ApoJ$^{-/-}$ mice (muscle: $n = 20$ per group, adipose tissue: $n = 16$ per group, BAT: $n = 20$ for control, $n = 19$ for L-ApoJ$^{-/-}$). Mice were studied at 18 weeks of age. **b** Insulin-stimulated glucose uptake in muscle, adipose tissue, and BAT was measured in control M-LRP2$^{-/-}$ mice (muscle: $n = 10$ for control, $n = 8$ for M-LRP2$^{-/-}$, adipose tissue: $n = 10$ for control, $n = 7$ for M-LRP2$^{-/-}$, BAT: $n = 10$ for control, $n = 8$ for M-LRP2$^{-/-}$). Mice were studied at 17 weeks of age. Mice were injected with insulin in combination with [$^{14}$C]2-deoxyglucose. In vivo 2-deoxyglucose uptake was measured. All bars and errors represent means ± SEM. $p$ values by two-sided Student's $t$-test in **a** and **b** are indicated.

muscle glucose metabolism[35,36]. One critical factor for this interorgan system is now identified as hepatokines from liver-derived proteins, which play a pivotal role in regulating glucose metabolism and insulin sensitivity in skeletal muscle[3,7]. Discovery of an unidentified hepatokine that is involved in insulin control of glucose metabolism in skeletal muscle is thus of particular interest. Here, we show that ApoJ functions as a hepatokine targeting insulin signaling and glucose metabolism in skeletal muscle; this action is mediated via the LRP2 signaling cascade. We propose a model for the involvement of the ApoJ → LRP2 signaling pathway in the maintenance of normal insulin signaling and glucose homeostasis.

We investigated the role of ApoJ in the regulation of glucose homeostasis and insulin signaling. Strikingly, our studies with animal models, including L-ApoJ$^{-/-}$ mice, global ApoJ$^{-/-}$ mice, and M-ApoJ$^{-/-}$ mice, demonstrated that the liver is a major source of circulating ApoJ and that liver-derived ApoJ appears in metabolically active organs, such as muscle, suggesting that ApoJ likely plays a role in interorgan communication networks. The fact that hepatic ApoJ deletion leads to impaired insulin action on muscle glucose uptake without any changes in body weight further suggests direct actions of liver-released ApoJ on insulin-dependent glucose metabolism. Given that LRP2 is found to be a potential receptor for ApoJ[20,22,37], it is conceivable that ApoJ released by the liver effects the action of insulin on glucose disposal in skeletal muscle via LRP2. Consistent with this, experimental evidence revealed that insulin's ability to stimulate glucose uptake is decreased in the absence of muscle LRP2. Thus, these findings suggest that ApoJ acts as a hepatokine that plays a pivotal role in modulating muscle glucose metabolism through LRP2, highlighting an interorgan communication network between liver and muscle.

The mechanisms for the involvement of LRP2 in the metabolic actions of insulin remain unknown to date. Here, we show that

insulin receptors on the cell surface of muscle cells failed to internalize in response to insulin when LRP2 expression was suppressed, suggesting that LRP2 is required for insulin-induced insulin receptor internalization, an early step in insulin signaling[38]. Correspondingly, phosphorylation of downstream signaling components for insulin, including IRS-1 and Akt, were greatly reduced in muscle of M-LRP2$^{-/-}$ mice. As a result, glucose uptake stimulated by insulin was decreased, ultimately leading to insulin resistance. Therefore, it seems likely that the cellular mechanism underlying insulin resistance induced by LRP2 deletion involves a major defect in the internalization of insulin receptor at the cell surface of muscle. Given that ApoJ is a ligand for LRP2[20,22,37], a similar mechanism for insulin resistance caused by a lack of circulating ApoJ was also detected, i.e., insulin's ability to activate insulin receptor was impaired in muscle of L-ApoJ$^{-/-}$ mice. However, we cannot rule out the possibility that ApoJ/LRP2-mediated insulin action could be independent of insulin receptor. Because LRP2 has NPXY motifs in the cytoplasmic tail[39], which are recognized by proteins containing a phosphotyrosine-binding domain, such as IRS[40,41], it is thus possible that ApoJ/LRP2 modulates PI3K signaling via an NPXY domain in a way that is independent of the insulin receptor. Indeed, ApoJ activates survival through the PI3K/Akt signaling pathway[42].

Interestingly, the current study found that while L-ApoJ$^{-/-}$ mice have no circulating ApoJ, M-LRP2$^{-/-}$ mice show a high level of circulating ApoJ, yet both are insulin resistant. These observations seem to be contradictory. However, a possible explanation for this is that defective insulin signaling in skeletal muscle of L-ApoJ$^{-/-}$ mice is likely due to impaired LRP2 signaling caused by a lack of its ligand ApoJ, which ultimately inhibits insulin action on glucose uptake. On the other hands, in M-LRP2$^{-/-}$ mice, circulating ApoJ could not act in muscle because of a lack of its cell surface receptor, thereby impairing insulin's ability to activate signaling. Under this condition, circulating ApoJ would remain high, as it cannot be utilized by muscle. In this regard, several studies show that elevated circulating ApoJ level is associated with insulin resistance in diet-induced obese mice[43], as well as humans with PCOS, type 2 diabetes, and atherosclerotic diseases[23,26]. However, insulin-sensitizing interventions, including exercise training[44,45] and pioglitazone treatment[46], lead to a significant decrease in circulating ApoJ in insulin-resistant humans. Thus, it is plausible that a certain level of ApoJ in the circulation may important in maintaining normal glucose homeostasis and insulin sensitivity. If beyond a certain threshold or a lack of ApoJ, ApoJ could desensitize insulin action on glucose metabolism. Together, we suggest that ApoJ signaling plays a key role as a co-factor in coordinating insulin-mediated signaling events, i.e., if ApoJ signaling is functionally absent, insulin signaling is affected correspondingly. Conversely, ApoJ signaling could be augmented if insulin signaling is improved, as in the cases of insulin-sensitizer treatment and exercise training[44,46]. Additional differences between L-ApoJ$^{-/-}$ and M-LRP2$^{-/-}$ mice include the fasting hyperglycemia and impaired GSIS seen in the former animals, suggesting a role for ApoJ in regulating ß-cell function that is retained in the presence of normal-to-elevated circulating ApoJ levels in the M-LRP2$^{-/-}$ mice. ApoJ regulation of insulin secretion represents a fertile topic for further study.

In this study, we further characterized the functional role of hepatic ApoJ in the regulation of hepatic lipid metabolism. We show herein that loss of hepatic ApoJ has no effects on hepatic lipid metabolism, including VLDL-TG production and hepatic lipid accumulation, indicating that metabolic dysfunctions caused by hepatic ApoJ deficiency is not likely due to changes in hepatic

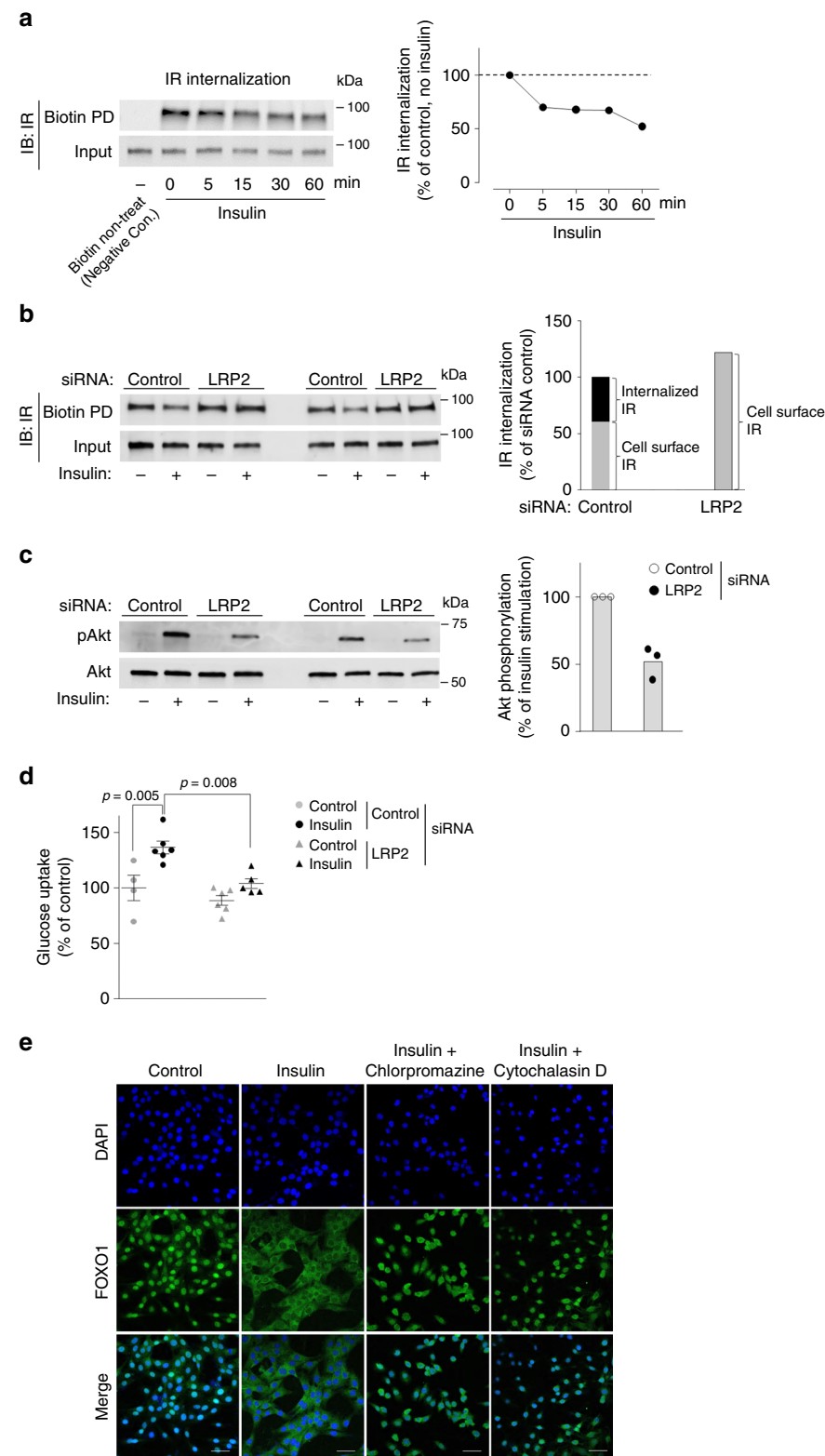

lipid metabolism. Moreover, in PCOS subjects, changes in muscle ApoJ by the insulin-sensitizer pioglitazone were observed in the absence of any alterations in lipoprotein levels. However, ApoJ is found to negatively regulate gene expression of SREBP1c, a master regulator of the lipogenic pathways, leading to reductions in hepatic lipid accumulation in HepG2 cells and diet-induced obese mice[47]. Similarly, liver-specific overexpression of ApoJ prevents the development of diet-induced hepatic steatosis[48],

suggesting a sufficiency of hepatic ApoJ in modulating hepatic lipid metabolism. Other studies revealed that treatment of HepG2 cells with recombinant ApoJ protein increased gene expression of key molecules involved in gluconeogenesis[49]. Thus, the importance of hepatic ApoJ in hepatic lipid metabolism will require further investigations.

Of note, a recent study demonstrated that ApoJ is secreted from human adipocytes in vitro and its production is increased in

**Fig. 7 Inhibition of LRP2 impairs insulin receptor internalization in $C_2C_{12}$ muscle cells. a** Time course of IR internalization following insulin binding in $C_2C_{12}$ muscle cells. Cells were serum starved for 4 h and incubated with insulin as indicated, followed by sulfo-NHS-Biotin treatment. The biotinylated surface protein fraction was pull-downed (PD) with streptavidin-agarose beads. Immunoblot indicates total IR in biotin-labeled cell surface (top) and total lysates (bottom). Line graph shows the quantification of the relative cell surface IR amount at different time points by insulin treatment. These data are representative from two independent experiments. **b** IR internalization following insulin binding in $C_2C_{12}$ muscle cells lacking LRP2. Cells were transiently transfected by siRNA. Cells were starved for 4 h and incubated with insulin for 15 min, followed by sulfo-NHS-Biotin treatment. The biotinylated surface protein fraction was PD by streptavidin-agarose beads. Immunoblot indicates total IR in biotin-labeled cell surface (top) and total lysates (bottom). Bar graph shows the quantification of the relative cell surface IR amount in cell surface IR and internalized IR. These data are representative from two independent experiments. **c** Insulin-stimulated Akt phosphorylation in $C_2C_{12}$ muscle cells lacking LRP2. Cells were transiently transfected by siRNA. Cells were starved for 4 h and incubated with insulin for 15 min. Total cell lysates were separated by SDS–PAGE. Akt was visualized by immunoblotting. Graph shows the quantification of insulin-stimulated Akt phosphorylation from three independent experiments. **d** Insulin-stimulated glucose uptake in in $C_2C_{12}$ muscle cells lacking LRP2. Cells were transiently transfected by LRP2 siRNA. Cells were starved for 3 h and incubated with insulin for 30 min; $n = 4$ for control (no insulin), $n = 6$ control (insulin), $n = 6$ for LRP2 siRNA (no insulin), $n = 5$ for LRP2 siRNA (insulin). [$^3$H]-2-deoxy-D-glucose uptake was measured. All bars and errors represent means ± SEM. $p$ values by one-way ANOVA followed by post hoc Tukey's HSD test are indicated. **e** Insulin-induced subcellular localization of FOXO1 in $C_2C_{12}$ cells. $C_2C_{12}$ muscle cells were treated with or without insulin in the presence of chlorpromazine or cytochalasin D. FOXO1 was detected by immunofluorescent analysis. These data are representative form three independent experiments. The scale bars represent 10 μm.

**Table 1 Clinical and metabolic characteristics of normal cycle and PCOS women.**

| | Normal cycle ($n = 7$) | PCOS-placebo ($n = 9$) | | | PCOS-pioglitazone ($n = 11$) | | |
|---|---|---|---|---|---|---|---|
| | | Pretreatment | Posttreatment | Change from pretreatment | Pretreatment | Posttreatment | Change from pretreatment |
| Age, years | 33.4 ± 2.3 | 27.5 ± 0.9* | – | – | 27.5 ± 1.3* | – | – |
| BMI, Kg/m² | 35.3 ± 2.5 | 34.8 ± 2.2 | 36.7 ± 2.4 | 1.9 ± 1.2 | 38.2 ± 1.8 | 38.9 ± 2.0† | 0.7 ± 0.6‡ |
| Fasting glucose, mg/dL | 99.3 ± 2.6 | 91.9 ± 1.3* | 92.0 ± 2.2 | 0.2 ± 2.4 | 96.1 ± 3.2 | 89.6 ± 2.1 | −6.5 ± 2.9 |
| Fasting insulin, IU/L | 14.3 ± 2.4 | 26.6 ± 2.7** | 27.7 ± 3.0 | 1.3 ± 1.6 | 36.3 ± 4.7** | 24.2 ± 4.2 | −12.2 ± 3.7‡ |
| HOMA2-IR | 1.9 ± 0.3 | 3.1 ± 0.3* | 3.5 ± 0.5 | 0.3 ± 0.3 | 4.4 ± 0.5** | 2.8 ± 0.5 | −1.6 ± 0.3# |
| HOMA-β, % | 116.4 ± 10.8 | 203.4 ± 16.1** | 210.0 ± 16.5 | 6.6 ± 12.5 | 240.4 ± 19.9** | 190.5 ± 17.6 | −49.9 ± 22.9 |
| GDR, mg/kg/min | 8.6 ± 0.7 | 7.1 ± 1.0 | 6.6 ± 0.9 | −0.5 ± 0.5 | 5.3 ± 0.5** | 6.5 ± 0.3 | 1.2 ± 0.3# |
| LDL cholesterol, mg/dL | 107.7 ± 16.9 | 116.4 ± 6.7 | 113.0 ± 10.2 | −3.4 ± 6.2 | 109.4 ± 6.9 | 110.9 ± 8.0 | 1.4 ± 8.5 |
| HDL cholesterol, mg/dL | 40.5 ± 1.9 | 44.8 ± 3.5 | 39.0 ± 2.2 | −5.8 ± 4.1 | 39.2 ± 2.9 | 41.7 ± 2.9 | 2.5 ± 3.0 |
| Triglyceride, mg/dL | 122.2 ± 16.8 | 102.1 ± 13.6 | 81.0 ± 11.6 | −21.1 ± 14.2 | 148.0 ± 13.9§ | 126.7 ± 21.2 | −21.3 ± 20.0 |
| FFA, mmol/L | – | 1.17 ± 0.11 | 1.06 ± 0.09 | −0.11 ± 0.19 | 1.15 ± 0.07 | 1.05 ± 0.10 | −0.10 ± 0.08 |
| Adiponectin, μg/mL | 17.7 ± 1.9 | 9.6 ± 1.5** | 11.4 ± 1.9 | 1.7 ± 2.7 | 9.1 ± 1.4** | 19.2 ± 2.3 | 10.1 ± 1.8# |

Data are expressed as frequency and means ± SEM. Comparison between groups was performed using ANOVA with a post hoc Fisher's PLSD test and paired Student's $t$-test.
BMI, body mass index; HOMA-IR, homeostasis model assessment-insulin resistance; HOMA-β (%), homeostasis model assessment-β (%); GDR, glucose disposal rate; FFA, free fatty acid.
*$p < 0.05$ vs normal cycle; **$p < 0.01$ vs normal cycle; §$p < 0.05$ vs PCOS-placebo; †$p < 0.05$ vs pretreatment; ‡$p < 0.05$ vs change from pretreatment of PCOS-placebo; #$p < 0.01$ vs change from pretreatment PCOS-placebo.

response to palmitate[49]. However, a lack of detailed experimental conditions is found in this report, even no information of ApoJ antibody is provided. The composition of the media for adipocyte culture is critical, because the level of ApoJ protein in serum is used for culture is very high[50,51], which could possibly effect the outcomes of these experiments. For example, it is possible that ApoJ in the media could be taken up by adipocytes, and then released from the adipocytes into the media in response to stimuli. This study only measured ApoJ level in the media but not adipocyte. Nevertheless, this observation should be confirmed in the context of in vivo milieu. From our studies with L-ApoJ$^{-/-}$ mice and global ApoJ$^{-/-}$ mice injected with an adenovirus encoding a secretory form of ApoJ, it is likely that ApoJ in adipose tissue could be that produced by the liver, as is the case with muscle ApoJ. However, this supposition would need to be confirmed by studying adipose-specific ApoJ$^{-/-}$ mice.

On the basis of our findings, we propose a key mechanism for involvement of ApoJ and LRP2 signaling in the metabolic action of insulin on glucose metabolism (Fig. 9); circulating ApoJ is secreted by the liver to enhance insulin-dependent muscle glucose uptake. Liver-derived ApoJ is delivered to skeletal muscle where it binds to its receptor LRP2 on the cell surface. ApoJ binding to LRP2 is a unique mechanism to amplify insulin action by specifically driving IR internalization. Thus, our study identifies ApoJ as a hepatokine that targets muscle LRP2 signaling, which plays a key role in regulation of glucose homeostasis and insulin

signaling. This ApoJ → LRP2 signaling axis can be added to the current networks of peripheral tissue-derived signals that serve as autocrine or paracrine signals in the regulation of whole-body fuel metabolism[7,8,52].

## Methods

**Animal care.** Animal studies were conducted in accordance with the National Institutes of Health "Guide for the Care and Use of Laboratory Animals" (NIH Publication No. 85-23, revised 1996) and approved by the Institutional Animal Care and Use Committees of Beth Israel Deaconess Medical Center. The mice were fed standard chow (Teklad F6 Rodent Diet 8664, Harlan Teklad, Indianapolis, IN) or a high-fat diet (HFD) with 58% kcal in fat (D12331, Research Diets, New Brunswick, NJ), and housed under controlled temperature at 22–24 °C and a 12 h light/12 h dark cycle with a humidity between 40–60%.

**Experimental animals.** Animals bearing a LoxP-flanked ApoJ allele (ApoJ$^{loxP/loxP}$ mice) were generated by inGenious Targeting Laboratory (Stony Brook, NY). Briefly, a BAC clone (C57BL/6, RPC123 clone) containing a 9.32 kb fragment of ApoJ genomic DNA was used to generate a targeting vector. Four independent ApoJ$^{loxP/+}$ ES clones were identified, which were injected into C57BL/6 blastocysts to generate chimeric mice. The chimeric mice were bred with wild-type C57BL/6 mice for germline transmission. Heterozygous animals were then crossed with mice expressing flp-recombinase in the germline (Flipper mice, from The Jackson Laboratory, Bar Harbor, ME) to delete the FRT-flanked Neo cassette. Offspring of these mice were heterozygous for the desired ApoJ$^{loxP/+}$ allele. Albumin-Cre; ApoJ$^{loxP/loxP}$ mice (L-ApoJ$^{-/-}$) and myogenin-Cre; ApoJ$^{loxP/loxP}$ (M-ApoJ$^{-/-}$) were generated by mating ApoJ$^{loxP/loxP}$ mice with albumin-Cre or myogenin-Cre transgenic mice, respectively. For generation of M-LRP2$^{-/-}$ mice, LRP2$^{loxP/loxP}$ mice[53] were crossed with myogenin-Cre transgenic mice. Genotypes of these mice were identified by polymerase chain reaction (PCR) or immunoblotting.

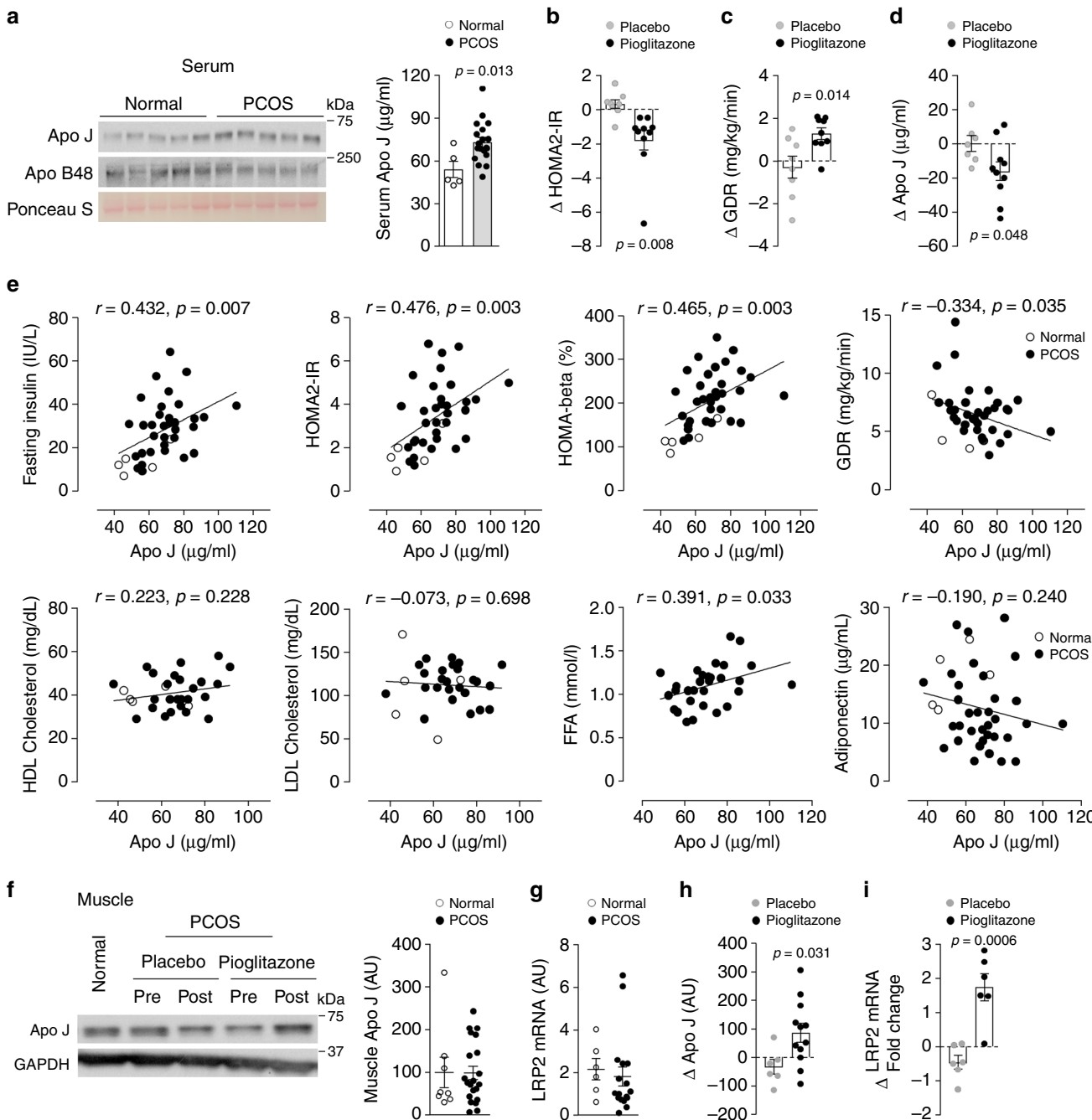

**Fig. 8 Serum ApoJ is elevated in insulin-resistant subjects with PCOS and normalized by pioglitazone, an insulin-sensitizing drug. a** Serum ApoJ levels in normal cycling (normal) and PCOS subjects. Immunoblots represent serum ApoJ and Apo B48 protein in normal and PCOS subjects. Ponceau S staining indicates that equal amounts of protein in each lane was loaded on the gel. Scatter diagram shows serum ApoJ levels measured by ELISA from normal ($n =$ 5) and PCOS subjects ($n = 19$). **b** Changes in HOMA2-IR after placebo ($n = 8$) or pioglitazone treatment ($n = 10$) in PCOS subjects. **c** Changes in glucose disposal rate after placebo ($n = 8$) or pioglitazone ($n = 9$) treatment in PCOS subjects. **d** Changes in serum ApoJ levels after placebo ($n = 7$) or pioglitazone treatment ($n = 10$) in PCOS subjects. All bars and errors represent means ± SEM. **e** Relationship of serum ApoJ with fasting insulin levels ($n =$ 38), HOMA2-IR ($n = 38$), HOMA-β (%) ($n = 38$), GDR ($n = 40$), HDL cholesterol ($n = 31$), LDL cholesterol ($n = 31$), free fatty acid ($n = 31$), and adiponectin ($n = 40$) levels in normal and PCOS subjects. $p$ values were obtained by Spearman's rank correlation analysis and $r$ values indicate Spearman's correlation coefficient. **f** Muscle ApoJ levels in normal ($n = 8$) and PCOS subjects ($n = 21$). PCOS subjects were treated with placebo or pioglitazone. **g** Muscle LRP2 levels in normal ($n = 6$) and PCOS subjects ($n = 19$). **h** Changes in muscle ApoJ levels after placebo ($n = 6$) or pioglitazone treatment ($n =$ 12) in PCOS subjects. **i** Changes in muscle LRP2 levels after placebo ($n = 6$) or pioglitazone ($n = 6$) treatment in PCOS subjects. All bars and errors represent means ± SEM. $p$ values by two-sided Student's $t$-test in **a**, **c**, **d**, **h**, and **i** are indicated.

Myogenin-Cre transgenic mice were generously provided by Dr. Zoltan Arany (University of Pennsylvania, Philadelphia, PA). Albumin-Cre transgenic mice and global ApoJ$^{-/-}$ mice were purchased from The Jackson Laboratory (Bar Harbor, Maine).

**Adenovirus-mediated gene transfer.** ApoJ$^{loxP/loxP}$ and L-ApoJ$^{-/-}$ mice were injected with a recombinant adenovirus encoding a secretory ApoJ at a concentration of $2 \times 10^9$ plaque-performing units (pfu) per gram of body weight via the tail vein. As a control, adenovirus encoding GFP was also injected. Blood

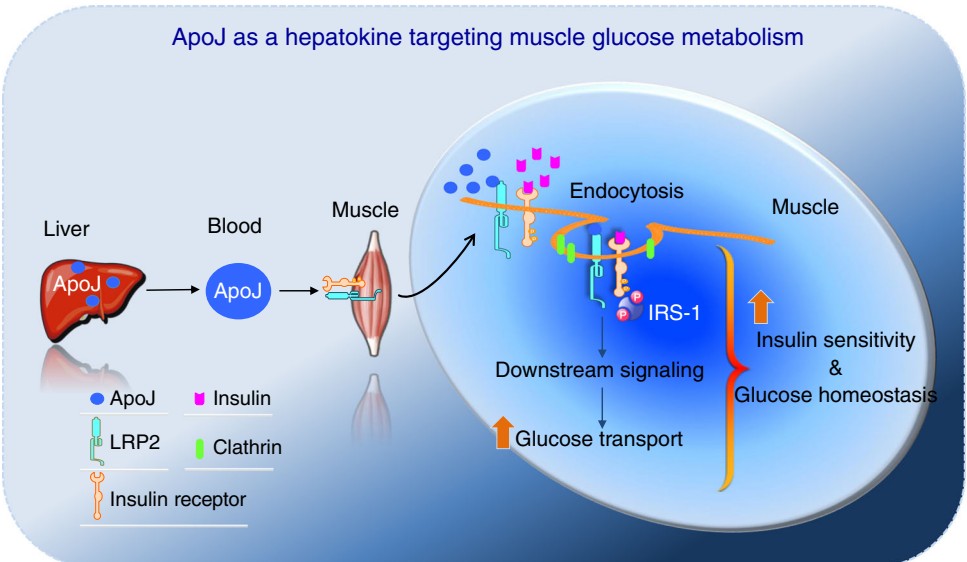

**Fig. 9 A role of the ApoJ → LRP2 signaling pathway in glucose homeostasis and insulin signaling.** Circulating ApoJ is dominantly produced by the liver. Liver-derived ApoJ then transports to skeletal muscle where it binds to its receptor LRP2 on the cell surface, leading to increased molecular interaction between LRP2 and IR. The complexes of LRP2 with the IR then undergo co-endocytosis, which is an essential step of insulin signaling in skeletal muscle. As a result, downstream signaling components activate, leading to enhanced glucose transport system activity. Thus, the ApoJ → LRP2 signaling pathway is central in regulation of glucose homeostasis and insulin signaling.

samples for analyzing serum ApoJ levels were drawn at 3–5 day after the adenovirus injections. Mice were sacrificed in the fasted state at 5 days after the injections. Tissues were rapidly removed, snap-frozen in liquid nitrogen, and stored at −80 °C until analysis.

**Tissue lysates**. Fifty milligrams of tissue were homogenized using a Tissue Lyser (Qiagen, Valencia, CA) in a 500 µl buffer A (20 mM Tris pH 7.5, 5 mM EDTA, 10 mM $Na_4P_2O_7$, 100 mM NaF, 2 mM $Na_3VO_4$) containing 1% NP-40, 1 mM PMSF, 10 µg/mL aprotinin, and 10 µg/mL leupeptin. Tissue lysates were solubilized by continuous stirring for 1 h at 4 °C, and centrifuged for 10 min at 14,000 × g. The supernatants were stored at −80 °C until analysis.

**RNA isolation and mRNA measurement**. Total RNA was isolated by the Trizol Reagent (Invitrogen, CA). Single strand cDNA from total RNA was synthesized by reverse transcription (RT-PCR) kit (Clontech, Mountain View, CA) according to the kit instruction. For ApoJ mRNA, PCR reaction was performed in a 20 µl volume containing 10 ng of cDNA for 35 cycles. The forward primer 5′-ACA ATG GCA TGG TCC TGG GAG AG-3′ and the reverse primer 5′-GTA TGC TTC AGG CAG GGC TTG C-3′ were used. The PCR products were separated on 0.9% agarose gels and analyzed with gel-imaging system (VersaDoc Multi Imaging Analyzer System, Bio-Rad Laboratories, Hercules, CA).

**Blood parameter measurements**. Blood was collected either from random-fed or overnight fasted mice via the tail. Blood glucose was measured using a OneTouch Ultra glucose meter (LifeScan, Inc., Milpitas, CA). Serum insulin and leptin levels were measured by ELISA (Crystal Chem, Chicago, IL). Serum total cholesterol and triglyceride levels were determined by enzymatic methods (Stanbio Laboratory, Boerne, TX). Serum IGF-1, serum IGFBP-1, plasma IGFBP-3, and serum ApoJ levels were determined by ELISA (MG100, DY1588-05, MGB300, MCLU00, R&D Systems, Minneapolis, MN). The range and sensitivity for these parameters are: glucose, 40–500 mg/dL; leptin, 1–25.6 ng/mL (sensitivity, 200 pg/mL using a 5-µL sample); insulin, 0.1–12.8 ng/mL (sensitivity, 0.1 ng/mL using a 5-µL sample); total cholesterol, 1–750 mg/dL; triglyceride, 1–1000 mg/dL; IGF-1, 31.2–2000 pg/mL (sensitivity, 8.4 pg/mL); IGFBP-1, 23.4–1500 pg/mL, IGFBP-3, 78.1–5000 pg/mL (sensitivity, 16 pg/mL); and ApoJ, 0.781–50 ng/mL (sensitivity, 0.031 ng/mL). Serum ALT and AST were measured by colorimetric assay kits (Cat# K752, Cat# K753, Biovision, Milpitas, CA) according to manufacturer's instruction. All assays were performed blinded.

**Body composition and energy expenditure**. Mice were weighted from 4 weeks of birth and weekly thereafter. Fat and lean body mass were assessed by EchoMRI (Echo Medical Systems, Houston, TX). Energy expenditure was measured by assessing oxygen consumption with indirect calorimetry. Individually housed male mice maintained on a chow diet until 8 weeks of age were studied using the Comprehensive Lab Animal Monitoring System (CLAMS, Columbus Instruments, Columbus, OH). Mice were acclimated in the CLAMS chambers for 72 h before

data collection. Mice had free access to food and water for the duration of the studies. During the course of the energy metabolism measurements ($O_2$ and $CO_2$) using CLAMS, high variations (overlapping) in measurements emerged at individual time points between groups. This did not allow us to statistically analyze individual time points. To enhance the statistical power of these measurements, we combined each value from the individual time points and analyzed the data by unpaired Student's $t$-tests to compare the two groups.

**GTT and ITT**. For glucose tolerance tests (GTT), male ApoJ$^{loxP/loxP}$ mice and L-ApoJ$^{−/−}$ mice at 13–14 weeks of age, or male LRP2$^{loxP/loxP}$ and M-LRP2$^{−/−}$ mice at 18 weeks of age were fasted overnight, and blood glucose was measured before and 15, 30, 60, 90, and 120 min after intraperitoneal injection of glucose (1.0 g/kg of body weight). For insulin tolerance tests (ITT), male ApoJ$^{loxP/loxP}$ mice and L-ApoJ$^{−/−}$ mice at 19–21 weeks of age, or male LRP2$^{loxP/loxP}$ and M-LRP2$^{−/−}$ mice at 24 weeks of age were fasted for 5 h, and blood glucose was measured before and 15, 30, 60, 90, and 120 min after intraperitoneal injection of human insulin (Humulin®, Lilly, Indianapolis, IN). A total of 0.5 unit/kg of body weight for mice fed a chow diet and 0.75 unit/kg of body weight for mice fed a HFD were used. Area under the curve for glucose or insulin was calculated[54].

**GSIS test**. For glucose-stimulated insulin secretion (GSIS) test, male ApoJ$^{loxP/loxP}$ mice and L-ApoJ$^{−/−}$ mice at 16–18 weeks of age, or male LRP2$^{loxP/loxP}$ and M-LRP2$^{−/−}$ mice at 22 weeks of age were fasted overnight, and serum insulin level was measured before and 15, 30, and 60 min after intraperitoneal injection of glucose (1.0 g/kg of body weight). Area under the curve for insulin was calculated[54].

**In vivo hepatic VLDL-triglyceride production**. Male ApoJ$^{loxP/loxP}$ mice and L-ApoJ$^{−/−}$ mice were injected with a 15% Tyloxapol solution in 0.9% NaCl (600 mg/kg; Sigma-Aldrich, St. Louis, MO) intraperitoneally after an overnight fast at 11 weeks of age. Triglyceride levels were measured before and 60, 120, 180, and 300 min after the injection.

**Liver histology and lipid assay**. The liver was fixed with 10% neutral buffered formalin (F5554, Sigma-Aldrich), embedded in paraffin, and stained with H&E. Liver lipids were extracted using chloroform/methanol (2:1) as a solvent. Total cholesterol and triglyceride content of the liver were determined by an enzymatic method (Cat#1010, Stanbio Laboratory; Cat #10010303, Cayman Chemical).

**In vivo glucose uptake**. Male mice at 17–18 weeks of age were studied. Twelve hours fasted mice were injected with either saline or insulin (2 mU/g) in combination with 0.33 µCi [$^{14}$C]2-deoxyglucose (2-DOG)/g and glucose (1 mg/g) administered via the retro-orbital sinus under isoflurane anesthesia. Mice were awake for the rest of procedure. Blood samples were taken 5, 10, and 15 min after injection for the determination of glucose and [$^{14}$C] levels to calculate the

integrated glucose-specific activity area under the curve. At 15 min, mice were sacrificed by decapitation and tissues rapidly collected and frozen in liquid nitrogen. Accumulation of $[^{14}C]2$-deoxyglucose into skeletal muscle, adipose tissue, and brown adipose tissue was measured using a perchloric acid and barium hydroxide/zinc sulfate $(Ba(OH)_2/ZnSO_4)$ precipitation procedure[55]. Tissues were weighed and homogenized in 0.5% perchloric acid. Homogenates were centrifuged and supernatants were neutralized with potassium hydroxide. One aliquot was counted directly to determine 2-DOG and phosphorylated 2-DOG (2-DOG-P) radioactivity. A second aliquot was treated with $Ba(OH)_2$ and $ZnSO_4$ to remove 2-DOG-P, and then counted to determine 2-DOG radioactivity. Glucose uptake was calculated from the difference between the radioactivity of these two aliquots divided by the integrated glucose-specific activity area under the curve and normalized to tissue weight.

**Acute insulin stimulation.** Male ApoJ$^{loxP/loxP}$ and L-ApoJ$^{-/-}$ mice at 24–26 weeks of age, or male LRP2$^{loxP/loxP}$ and M-LRP2$^{-/-}$ mice at 24 weeks of age were fasted overnight. Mice were injected intraperitoneally with human insulin (10 U/kg of body weight; Humulin®) or saline and scarified 10 min later. Tissues were rapidly removed, snap-frozen in liquid nitrogen, and stored at $-80\,°C$ until analysis.

**Immunoblotting analysis.** Tissue lysates (20–50 µg protein) or cell lysates (20 µg protein) were resolved by sodium dodecyl sulfate–polyacrylamide gel electrophoresis (SDS–PAGE) and transferred to nitrocellulose membranes (GE Healthcare Life Sciences, Pittsburgh, PA). The membranes were incubated with: polyclonal antibodies against phosphor-Y$^{972}$ IR (Cat#: 44-800 G); phosphor-Y$^{612}$ IRS-1 (Cat#: 44-816 G; both Invitrogen; Carlsbad, CA); phospho-Ser$^{473}$Akt (Cat#: 4060); phospho-Thr$^{308}$Akt (Cat#: 4056); phospho-Thr$^{642}$AS160 (Cat#: 4288); phospho-Ser$^{21/9}$ GSK3 (Cat#: 9331); phospho-Ser$^{641}$ glycogen synthase (Cat#: 3891); phosphor-Ser$^{240/244}$ S6 ribosomal protein (Cat#: 2215; all Cell Signaling Technology, Beverly, MA); active MAPK (pTEpY) antibody (Cat#; V8031; Promega, Madison, WI); ApoJ/Clusterin (sc-8354); IR (sc-711); Akt (sc-7126); GAPDH (sc-47724); selenoprotein P (sc-376858; all Santa Cruz Biotechnology, Dallas, TX), or monoclonal antibodies against ß-actin (A2228, Sigma-Aldrich, St. Louis, MO). Dilution of all antibodies was 1:1000 except active MAPK, which was 1:5000. The membranes were washed with Tris-buffered saline (TBS) containing 0.05% Tween 20 for 30 min, incubated with horseradish peroxidase secondary antibodies (1:2000 dilution; GE Healthcare Life Sciences) for 1 h, and washed with TBS containing 0.05% Tween 20 for 30 min. The bands were visualized with enhanced chemiluminescence and quantified by an ImageJ program (v1.52a, NIH). All phosphorylation data were normalized by total protein levels. Uncropped and unprocessed scans of all blots are provided in the Source Data File.

**Immunoprecipitation analysis.** For IRS-2 phosphorylation, liver lysates (1 mg) were subjected to immunoprecipitation with 5 µl of a polyclonal IRS-2 antibody (a gift from Dr. Morris White, Children's Hospital, Boston, MA) coupled with protein A-Sepharose (Sigma-Aldrich). The immunoprecipitates were washed three times with buffer A, and resuspended in a 4× Laemmli sample buffer, and heated for 5 min. The immunoprecipitates were resolved by SDS–PAGE and transferred to nitrocellulose membranes. The membranes were incubated with a monoclonal antibody against phosphotyrosine (PY 20; Santa Cruz Biotechnology). The bands were visualized with enhanced chemiluminescence and quantified by an ImageJ program.

**In situ PLA.** Male LRP2$^{loxP/loxP}$ and M-LRP2$^{-/-}$ mice at 18 weeks of age were fasted overnight. Gastrocnemius muscles were harvested and fixed with formalin overnight and embedded with paraffin. Muscle sections (5 µm) were boiled in 10 mM citrate buffer for antigen retrieval followed by blocking at 37 °C and then incubated with primary antibody against LRP2 (Cat#; NB110-96417, Novus biologicals, Littleton, CO) overnight. PLA was performed using the Duolink® In Situ Detection Reagents Red with PLA probes Anti-PLUS and Anti-MINUS for mouse (Sigma-Aldrich). The nuclei of cells were stained using Duolink® In Situ Mounting Medium with DAPI (Sigma-Aldrich). Images were captured by a confocal microscope (Zeiss LSM 880, Zeiss). For in vitro studies, C$_2$C$_{12}$-myc-GLUT4 cells were incubated with 100 nM ApoJ (a gift from Dr. Min Bon Hong, Korea University, Seoul, Korea) or 100 nM insulin for 15 min. Cells were washed ice-cold PBS three times and fixed with ice-cold methanol. Cells were incubated with primary antibodies against ApoJ (Cat#; sc-8354, Santa Cruz Biotechnology) and LRP2 (Novus biologicals) overnight. PLA was performed using the Duolink® In Situ Detection Reagents Red with Duolink® In Situ PLA probe Anti-rabbit PLUS and Anti-mouse MINUS (Sigma-Aldrich). The nuclei of cells were stained using Duolink® In Situ Mounting Medium with DAPI (Sigma-Aldrich). Images were captured by a fluorescence microscope (Leica DMi8, Leica) and image data were analyzed by AxioVision software (v 4.2 SP1, ZEISS).

**Cell culture.** C$_2$C$_{12}$ cells (ATCC, CRL-1772) were maintained in high glucose DMEM (Thermo fisher scientific, Waltham, MA) supplemented with 10% FBS (Thermo fisher scientific) and 1% Antibiotic-Antimycotic (Thermo fisher scientific). C$_2$C$_{12}$-myc-GLUT4 cells (a gift from Dr. Amira Klip, The Hospital for Sick Children, Toronto, Canada) were maintained in high glucose DMEM supplemented with 10%

FBS, 1% Antibiotic-Antimycotic and 100 µg/mL Blasticidin S HCL (Thermo fisher scientific). Cell lines were tested monthly for mycoplasma contamination.

**Transfection.** C$_2$C$_{12}$-myc-GLUT4 cells were plated at a density of $1 \times 10^5$/well on six-well plates on the day of transfection. Lipofectamine RNAiMax (Thermo fisher scientific), 100 nM Lrp2 siRNA (s201412, Thermo fisher scientific), or scRNA (negative control siRNA, AM4635, Thermo fisher scientific) was diluted in media, respectively. The diluted siRNAs were added to the diluted Lipofectamine RNAiMax, and then incubated for 15 min at room temperature. The siRNA–Lipofectamine complex was added to the cells. After transfection for 48 h, the transfected cells were subjected to the isolation of total RNA or protein.

**Insulin receptor internalization assay.** The LRP2 siRNA- or scRNA-transfected C$_2$C$_{12}$-myc-GLUT4 cells were serum starved for 4 h, followed by the stimulation with 100 nM insulin for 0–30 min. The cells were rinsed with ice-cold PBS, followed by 0.3 mg/mL sulfo-NHS-Biotin (Thermo fisher scientific) labeling at 4 °C for 30 min. Then, the cells were quenched with ice-cold 100 mM glycine at 4 °C for 10 min. The cells were washed three times with ice-cold PBS, and then lysed with NP40 Cell Lysis Buffer (Thermo fisher scientific) supplemented with Halt™ Protease and Phosphatase Inhibitor Cocktail (Thermo fisher scientific) and 1 mM PMSF. The 100 µg total protein was incubated with 10 µl streptavidin-agarose beads (50% slurry, Thermo fisher scientific) at 4 °C overnight to pull down the biotinylated surface proteins. The beads were washed with lysis buffer three times, and then boiled in SDS–PAGE loading buffer (NuPAGE™ LDS Sample Buffer containing with NuPAGE™ Sample Reducing Agent (Thermo fisher scientific) at 100 °C for 10 min. The biotinylated surface protein fraction and the total protein lysate were resolved on 4–16% Mini-PROTEAN TGX™ precast protein gels (Bio-Rad, Hercules, California), transferred to nitrocellulose membrane, and immunoblotted using each antibody. The bands were visualized with enhanced chemiluminescence and quantified by an ImageJ program.

**In vitro glucose uptake.** C$_2$C$_{12}$-myc-GLUT4 cells were serum starved for 3 h in serum-free DMEM (25 mM glucose) at 37 °C. and then washed once with glucose-free minimum Eagle's medium. Insulin (100 nM) or vehicle was added for 30 min, followed by the addition of 100 µM $[^3H]$-2-deoxy-D-glucose (PerkinElmer, Waltham, MA), 0.33 µCi per 35-mm diameter well. Cells were incubated for a further 10 min at 37 °C. Transport was subsequently stopped by placing the cells on ice and adding 1:1 (v/v) ice-cold phloretin solution (82 µg/liter in phosphate-buffered saline; Sigma). Cells were washed once with cold PBS, solubilized in 1 N NaOH at 37 °C for 30 min, and $[^3H]2$-deoxy-D-glucose incorporation was measured by liquid scintillation counting. The results were expressed as a percentage of basal glucose transport in control cells.

**Immunofluorescent analysis.** C$_2$C$_{12}$ cells were treated with or without insulin (10 nM) in the presence of chlorpromazine (30 µM) or cytochalasin D (2 µM). Cells were then fixed in methanol and washed with PBS. The fixed cells were incubated overnight with a rabbit antibody to FOXO1 (ab70382, 1:50, Abcam, Cambridge, MA) at 4 °C overnight. The primary antibody was washed and slides were incubated for 1 h with an Alexa Fluor 488-conjugated antibody to rabbit IgG (Cat#; A-11008, 1:500, Invitrogen, Carlsbad, CA). For counterstaining of nuclei, cells were incubated with DAPI (Sigma-Aldrich). Images of green fluorescence were assessed by a confocal microscope (Zeiss LSM 880, Zeiss).

**Quantitative real-time PCR.** Total RNA was extracted from each tissue or cell using a TRIzol reagent (Invitrogen, CA) and subjected to quantitative real-time PCR as described[56]. Single stranded cDNA from the total RNA was synthesized with a RT-PCR kit (Clontech, Mountain View, CA) according to the kit's instructions. Quantitative RT-PCR was performed with an Applied Biosystems 7900HT Fast System using either a SYBR Green PCR Mastermix reagent (Applied Biosystems, Foster City, CA) or TagMan® Gene Expression Assays (Applied Biosystems). Relative mRNA expression levels were determined using the $2^{-\Delta\Delta CT}$ method normalized to 36B4. For LRP2 mRNA in human muscle, the forward primer 5′-CCA GCA AGG AAC CAG AGA ACA-3′ and the reverse primer 5′-AGG CAG AGC AAA GCA GAG ATG-3′ were used. For LRP2 mRNA in C$_2$C$_{12}$ cells, the forward primer 5′-GAG TGC ATC CTT CGT GCC TAT-3′ and the reverse primer 5′-CAG CCA TCC TCA TCA CCA GAA-3′ were used. For LRP2 mRNA in mouse muscle, the commercial primers for LRP2 (Mm01328171_m) and 18 s (Mm0427787_s1) (Applied Biosystems) were used.

**Clinical study.** The current study is a post hoc analysis of samples collected from the parent study under which subjects were recruited and treated which has been described previously[57,58]. The experimental protocol and post hoc analysis were approved by the Human Research Protection Program of the University of California, San Diego and VASDHS IRB. Our studies were performed prior to ClinicalTrials.gov identifier requirements. Twenty-eight PCOS and six non-PCOS subjects were recruited between March 20, 2002 and January 6, 2006. Informed written consent was obtained from all subjects after explanation of the protocol. We have complied with all relevant ethical regulations regarding studies with human research participants.

Briefly, subjects with PCOS were screened by medical history, physical examination, and laboratory evaluation. The diagnosis of PCOS was based on criteria recommended by the 1990 National Institutes of Health Conference on PCOS[59]. All participants were pre-menopausal, and medication use was stable for at least 60 days before screening. None of the subjects were diabetic or had a family history of type 2 diabetes. Other exclusion criteria included pregnancy; weight loss or gain >3 kg over the last 3 months; a basal 17-OHP value of <3 ng/mL (9.1 nmol/L); or medication use, including glucocorticoids, antiandrogens, ovulation induction agents, metformin, and anti-obesity agents within 60 days prior to screening. All procedures were performed in the early or mid-follicular phase (d 2–8) of the subjects' menstrual cycle, except in those subjects who did not have regular menses. Following pretreatment testing, including glucose clamp and percutaneous needle biopsies of vastus lateralis muscle before clamp steady-state insulin infusion, subjects with PCOS were given either placebo or pioglitazone, 45 mg/d, for 6 months. At the end of the treatment period, each subject underwent the same tests as in the pretreatment phase. All studies were performed after a 12–14 h overnight fast. Blood was collected and serum isolated and immediately stored at −80 °C. Insulin action in the fasting state was determined in all subjects by calculating the HOMA2-IR[60]. All of the subjects had maximal insulin action measured by a 3 h hyperinsulinemic (300 mU/m$^2$/min) euglycemic (5.0–5.5 mmol/L) glucose clamp; the GDR was measured during the last 30 min of the clamp steady-state period[61]. All of the prespecified primary outcomes of the parent study have been reported previously (57) and there were no predefined secondary outcomes in the study protocol. ApoJ levels were measured by ELISA kit (Boster biological technology, Pleasanton, CA, USA; sensitivity = 20 pg/mL, interassay cv = 6.9–7.5%, intraassay cv = 4.2–4.6%). Samples were diluted to be analyzed within range of the standard curve (0.78–50 ng/mL). Some samples of ApoJ levels were also assessed by immunoblotting analysis. Adiponectin levels were measured by radioimmunoassay kit from Linco (St. Louis, MO; sensitivity = 2 μg/mL, interassay cv = 7%, intraassay cv = 11%). Free fatty acid levels were measured by colorimetric assay (Wako Chemicals, Nuess, Germany). TG, HDL cholesterol, and LDL cholesterol were measured by quantitative enzymatic assay at ARUP Laboratories (Salt Lake City, Utah).

**Statistical analyses**. Data are presented as means ± SEM and individual data points are plotted. Sample size was determined by our experience with inherent variability, especially with outbred strains. No statistical method was used to predetermine sample size. The distribution of the data and homogeneity of variances were analyzed by Shapiro–Wilk and Levene's tests, respectively. Unpaired Student's t-tests were used throughout this study to compare two distinct groups if the data were distributed normally, unless non-parametric Mann–Whitney $U$ test was used. When more than two groups were compared, one-way analysis of variance (ANOVA) was performed with post hoc tests, Fisher's PLSD and Tukey HSD tests. Repeated measures two-way ANOVA was performed for GTT, ITT, and GSIS. When intervention or interaction (intervention-by-time) was significant by repeated measures two-way ANOVA, post hoc analyses were performed by providing extra codes into the SPSS program for multiple comparisons. The paired t-test was used to compare pretreatment- and posttreatment. Statistical analyses were performed using Stat View version 5.0 (Abacus Concepts, Berkeley, CA) and SPSS version 18.0 for window (SPSS, Inc., Chicago, IL). Spearman's rank correlation analysis was used to assess the correlations between ApoJ levels and each covariate was evaluated. All reported $p$ values were two-sided unless otherwise described. Differences were considered significant at $p < 0.05$.

**Reporting summary**. Further information on research design is available in the Nature Research Reporting Summary linked to this article.

## Data availability

The authors declare that the data supporting the findings of this study are available within the paper and its supplementary information files. Source data underlying Figs. 4a–j, l–o, 5a, b, d–i, 6a, b, 7a–d, and 8a–i, and Supplementary Figs. 2a–l, 3a–h, 4b–d, and 5 are provided as a Source Data File. The numerical data underlying Table 1 is provided as a Source Data File.

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

## Acknowledgements

This work was supported by grants from the National Institutes of Health (R01DK111529 and R01DK106076 to Y.-B.K.) and in part by Merit Review Award (I01CX00635) from the United States Department of Veterans Affairs Clinical Sciences Research and Development Service (R.R.H.), grants from the Basic Science Research Program through the National Research Foundation of Korea (NRF) funded by the Ministry of Education (2018R1D1A1B07049123 to J.A.S. and 2017R1A6A3A03003298 to W.-M.Y.), a grant from the Korean Diabetes Association (2017S-2 to J.A.S.), and a grant from Korea University (K1813091 to J.A.S.). The contents do not represent the views of the U.S. Department of Veterans Affairs or the United States Government. The Animal Metabolic Physiology Core (P30 DK057521 Barbara Kahn) performed in vivo glucose uptake. M.-C.K. is a recipient of a post-doctoral fellowship award from the American Diabetes Association (1-17-PDF-146), I.S.L. is a recipient of FCT fellowship from Portugal (SFRH/BD/71021/2010), and L.P.M is a recipient of São Paulo Research Foundation from Brazil (FAPESP 2013/14149-6). We would like to thank Barbara Kahn, Tony Hollenberg, and Terry Flier for helpful discussions, Odile Peroni for technical assistance on in vivo glucose uptake, Amira Klip for the C2C12-myc-Glut4 cell line, Sungman Cho for glucose uptake assays, Jin Sung Park for VLDL-secretion, Wendy Li for Immunofluorescent analysis, Zoltan Arany for myogenin-Cre transgenic mice, Inkyu Lee for ApoJ adenovirus, and Min Bon Hong for ApoJ recombinant protein.

## Author contributions

J.A.S., M.-C.K., and Y.-B.K. designed the study. J.A.S and M.-C.K. performed most of experiments with L-ApoJ$^{-/-}$, M-ApoJ$^{-/-}$, and M-LRP2$^{-/-}$. J.A.S. and M.-C.K. performed ApoJ adenovirus study. W.-M.Y., K.S.P., and M.S.K performed in vitro experiments. W.-M.Y. and A.U. performed PLA assay. J.A.S. and J.-I.H. measured serum metabolic parameters. S.S.K. and W.M.H. measured ApoJ levels in human samples and analyzed human data. A.V. and M.-C.K. performed in vivo glucose uptake. H.H., S.H.L., and I.S.L. carried out the genotyping of the experimental mice. Y.D., S.H.H., and L.P.M. carried out immunoblotting analysis. T.E.W. produced LRP2 floxed mice. V.A., R.R.H., and T.P.C. conducted the clinical study and collection of human samples, and contributed to the editing of the manuscript. All authors discussed the results and commented on the manuscript. J.A.S. and Y.-B.K. wrote the manuscript.

## Competing interests

The authors declare no competing interests.
