## [Peer Review File · Nature Communications]

Reviewers' Comments:

Reviewer #1:

Remarks to the Author:

The authors of this article claim that ApoJ is a novel "hepatokine" that regulates muscle glucose metabolism via effects on insulin action. The physiological evidence for this claim is based on studies using a liver-specific ApoJ knockout, and on studies using an muscle-specific LRP2 knockout. Evidence for the claim that LRP2 is necessary for IR endocytosis comes from studies using C2C12 cells.

This is an interesting and novel theory of function for ApoJ that, if true, would certainly have a high impact on the field. Liver-specific ApoJ knockouts are shown to have a metabolic phenotype of impaired glucose tolerance. To my eye, the insulin sensitivity data shown in Fig. 2D are not convincing in suggesting impaired glucose clearance following insulin injection. The authors do report differences in the activity of the IR tyrosine kinase and subsequent signaling cascade(s). Muscle-specific LRP2 (and liver-specific ApoJ) knockout mice also exhibit evidence of impaired insulin action in muscle with respect to glucose uptake (Fig. 3A, B). The authors also claim that LRP2 works by actions involving IR endocytosis; the model reported is that ApoJ facilitates interaction of LRP2 with the IR, and that this step is required for IR endocytosis.

Is the work convincing, and if not, what further evidence would be required to strengthen the conclusions?

While I am impressed with the data presented, in my opinion the experimental evidence falls short of proving the mechanism proposed to explain the observations. I also feel that much has been left out of the description of the function of ApoJ in the 2nd paragraph of the Introduction. The authors cite a series of studies that support their claim, but don't discuss other findings that could raise red flags for the reviewer. ApoJ is a chaperone that aids the folding of secreted proteins. It has three isoforms, which have been differentially implicated in pro- or antiapoptotic processes.

- 1) ApoJ is a lipoprotein - its role as an apolipoprotein has not been adequately addressed. The associations between serum ApoJ and insulin resistance in mice and humans could also reflect differences in hepatic VLDL output.
- 2) ApoJ/Clusterin is also a chaperone involved in protein folding - how has loss of ApoJ affected hepatic function? Is VLDL production normal? Is the secretion of other secreted proteins (e.g., IGF1, IGF1BP etc.) normal? Are the livers "healthy" or steatotic? Does secretion of ApoJ correlate with increased VLDL production in obesity?
- 3) The claim that IR endocytosis is involved in insulin signaling is not novel (Mol Cell Biol. 1998 Jul; 18(7): 3862-3870). However, the authors present very little data supporting the concept that LRP2 interacts with the IR. I am not actually sure what Fig.3C and D are telling us, as I suspect all it shows is less insulin binding in skeletal muscle. More evidence for a direct interaction of the IR with LRP2 is needed, and more data on how the lack of LRP2 affect IR internalization is also required.
- 4) Similarly, there is scant evidence for the claim that muscle-LRP2 knockout mice are "ApoJ resistant". If ApoJ is part of the lipoprotein complex, and its interaction with LRP2 is needed for uptake of VLDL in muscle, then one could contemplate reduced clearance. There is also a need for some kind of feedback mechanism between muscle and the liver to explain how the liver is sensing ApoJ action to compensate. For insulin, failure to adequately increase glucose disposal leads to higher glucose, which then leads to more glucose-stimulated insulin secretion. I am neither convinced nor impressed by this paragraph. Also, the statement on how crossing muscle-LRP2 and

liver-ApoJ knockout mice will restore "ApoJ sensitivity" is pointless, and invites a reviewer to request that specific experiment be done. I am not, as I am not convinced it will address the underlying principle.

5) Treatment of PCOS patients with a glitazone could indirectly affect ApoJ by effects on hepatic lipoprotein metabolism.

Reviewer #2:

Remarks to the Author:

In this manuscript entitled "ApoJ is a novel hepatokine regulating muscle glucose metabolism and insulin sensitivity," the authors investigated the role of apolipoprotein J (ApoJ) on insulin resistance and glucose intolerance in a low-density lipoprotein receptor-related protein-2 (LRP2)-dependent manner. The authors stated the importance of the ApoJ and LRP2-insulin (IR) axis. This study includes interesting data; however, there are numerous inadequate description and over-interpretation, particularly the relation to LRP2 is suspicious. In this manuscript, the authors showed neither the protein levels of LRP2 nor the direct bound between ApoJ and LRP2. It is also unclear that LRP2 is expressed in skeletal muscle. The authors should show the role of LRP2 in vitro experiments using C2C12 myocytes. Previous reports showed that the expression levels of LRP2 in skeletal muscle are quite low. Therefore this reviewer felt the possibility that the statements include over-interpretation. Other comments are as follows:

Major comments:

1. The abbreviation "ApoJ" is not suitable in the title. In addition, I could not find this description in the abstract.
2. In the introduction, the description of other hepatokine is lacked. Selenoprotein P is known as a hepatokine, and one of its receptor is LRP2. The authors should mention about this hepatokine. I'm wondering that the effects of ApoJ are dependent or independent of selenoprotein P.
3. In the results section, the authors stated "circulating ApoJ is retained in muscle via LRP2 and that LRP2 signaling is essential for maintenance of ApoJ homeostasis in circulation." It is unclear whether KO of LRP2 is specific in skeletal muscle. The authors should demonstrate this in the LRP2 protein levels. In addition, the authors should show the co-localization of ApoJ with LRP2 by using immunohistochemical method. It is also important to prove this statement by using in vitro experiments.
4. Western blot analysis of ApoJ is plausible because the molecular weight of several blots is not constant.

Overview of revision: We would like to thank the reviewers for their constructive comments, which have contributed to a more mechanistic study of the manuscript. We addressed most of comments by performing additional experiments. New data are below;

Fig. 4d: Serum IGF1 level

Fig. 4e: Serum IGFBP-1 level

Fig. 4f: Serum IGFBP-3 level

Fig. 4g: Serum TG level

Fig. 4h: Serum cholesterol level

Fig. 4i: Hepatic TG level

Fig. 4j: Hepatic cholesterol level

Fig. 4k: Hepatic H&E staining

Fig. 4l: Hepatic VLDL production

Fig. 5i: Glucose-induced insulin secretion

Fig. 7a-b: Insulin-stimulated IR internalization in C₂C₁₂ cells

Fig. 7c: Akt phosphorylation in C₂C₁₂ cells

Fig. 7d: Glucose uptake in C₂C₁₂ cells

Fig. 8e: Correlations between ApoJ and HDL, LDL, FFA, or adiponectin in humans

Table 1: Clinical and metabolic characteristics of normal cycle and PCOS women

Sup Fig. 2f: Selenoprotein P level in serum

Sup Fig. 2g: Selenoprotein P level in the liver

Sup Fig. 2h: Hepatic gene expression involved in glucose metabolism

Sup Fig. 2i: Muscle LRP2 mRNA level

Sup Fig. 4b: LRP2 gene expression in multiple organs

Sup Fig. 4d: LRP2 gene expression in C₂C₁₂ cells transfected with siLRP2

Sup Fig. 4e: LRP2 expression in muscle

Sup Fig. 4f: Co-localization of ApoJ and LRP2 in muscle

Sup Fig. 4g: Co-localization of ApoJ and LRP2 in C₂C₁₂ cells

Sup Fig. 5: Correlations between ApoJ and age, BMI, or TG in humans

Response to the Reviewers

Reviewer #1 (Remarks to the Author):

The authors of this article claim that ApoJ is a novel “hepatokine” that regulates muscle glucose metabolism via effects on insulin action. The physiological evidence for this claim is based on studies using a liver-specific ApoJ knockout, and on studies using an muscle-specific LRP2 knockout. Evidence for the claim that LRP2 is necessary for IR endocytosis comes from studies using C2C12 cells. This is an interesting and novel theory of function for ApoJ that, if true, would certainly have a high impact on the field. Liver-specific ApoJ knockouts are shown to have a metabolic phenotype of impaired glucose tolerance. To my eye, the insulin sensitivity data shown in Fig. 2D are not convincing in suggesting impaired glucose clearance following insulin injection. The authors do report differences in the activity of the IR tyrosine kinase and subsequent signaling cascade(s). Muscle-specific LRP2 (and liver-specific ApoJ) knockout mice also exhibit evidence of impaired insulin action in muscle with respect to glucose uptake (Fig. 3A, B). The authors also claim that LRP2 works by actions involving IR endocytosis; the model reported is that ApoJ facilitates interaction of LRP2 with the IR, and that this step is required for IR endocytosis. Is the work convincing, and if not, what further evidence would be required to strengthen the conclusions? While I am impressed with the data presented, in my opinion the experimental evidence falls short

of proving the mechanism proposed to explain the observations. I also feel that much has been left out of the description of the function of ApoJ in the 2nd paragraph of the Introduction. The authors cite a series of studies that support their claim, but don't discuss other findings that could raise red flags for the reviewer. ApoJ is a chaperone that aids the folding of secreted proteins. It has three isoforms, which have been differentially implicated in pro- or antiapoptotic processes.

Response: We thank the reviewer for instructive suggestions. As the reviewer suggested, we have now expanded the description of ApoJ's role in the context of chaperone function in the 2nd paragraph of the introduction. Please see our point-by-point responses below.

1) ApoJ is a lipoprotein - its role as an apolipoprotein has not been adequately addressed. The associations between serum ApoJ and insulin resistance in mice and humans could also reflect differences in hepatic VLDL output.

Response: As the reviewer requested, we measured hepatic VLDL production in control and liver-specific ApoJ-deficient (L-ApoJ^{-/-}) mice. We found that hepatic VLDL production was not different between control and (L-ApoJ^{-/-}) mice. These data are now presented in Fig. 4l. The results are described on page 6, in the 1st paragraph.

2) ApoJ/Clusterin is also a chaperone involved in protein folding - how has loss of ApoJ affected hepatic function? Is VLDL production normal? Is the secretion of other secreted proteins (e.g., IGF1, IGFBP etc.) normal? Are the livers "healthy" or steatotic? Does secretion of ApoJ correlate with increased VLDL production in obesity?

Response: As the reviewer suggested, we have performed additional experiments to measure VLDL production, serum IGF1, serum IGFBP-1, serum IGFBP-3, serum and hepatic lipid profiles (TG and cholesterol) and H&E staining. We found that serum IGFBP-1 levels were significantly increased in L-ApoJ^{-/-} mice compared with control mice. These data are now presented in Fig. 4e. The results are described on page 6, in the 1st paragraph. However, serum IGF1, serum IGFBP-3, serum TG and cholesterol, hepatic TG and cholesterol as well as hepatic H&E staining did not differ between control and L-ApoJ^{-/-} mice. These data are now presented in Fig. 4d, 4f, 4g, 4h, 4i, 4j, and 4k, respectively. The results are described on page 6, in the 1st paragraph.

Does secretion of ApoJ correlate with increased VLDL production in obesity?

Response: As shown in Fig. 4l, hepatic ApoJ knockout had no effect on VLDL production. Furthermore, our studies in obese PCOS subjects indicate that ApoJ was not correlated with either HDL or LDL levels (Fig. 8e, page 11, 1st paragraph). Taken together, this evidence leads us to think that ApoJ may not be correlated with VLDL production in obesity.

3) The claim that IR endocytosis is involved in insulin signaling is not novel (Mol Cell Biol. 1998 Jul; 18(7): 3862–3870). However, the authors present very little data supporting the concept that LRP2 interacts with the IR. I am not actually sure what Fig.3C and D are telling us, as I suspect all it shows is less insulin binding in skeletal muscle. More evidence for a direct interaction of the IR with LRP2 is needed, and more data on how the lack of LRP2 affect IR internalization is also required.

Response: We thank the reviewer for these important points. We have removed the interaction data of IR with LRP2 from the manuscript as our current data are premature, as the reviewer pointed out.

According to the reviewer's suggestion, we have performed additional experiments to determine whether LRP2 action affects IR internalization. We found that insulin stimulated IR internalization by ~40% but this effect was abolished by LRP2 inhibition in C₂C₁₂-myc-Glut4 cells, suggesting that LRP2 is required for insulin-induced insulin receptor internalization. These results could explain the key mechanism underlying ApoJ/LRP2 deletion-induced insulin resistance. These data are now presented in Fig. 7b. The results are described on page 10, in the 2nd paragraph.

4) Similarly, there is scant evidence for the claim that muscle-LRP2 knockout mice are "ApoJ resistant". If ApoJ is part of the lipoprotein complex, and its interaction with LRP2 is needed for uptake of VLDL in muscle, then one could contemplate reduced clearance. There is also a need for some kind of feedback mechanism between muscle and the liver to explain how the liver is sensing ApoJ action to compensate. For insulin, failure to adequately increase glucose disposal leads to higher glucose, which then leads to more glucose-stimulated insulin secretion. I am neither convinced nor impressed by this paragraph. Also, the statement on how crossing muscle-LRP2 and liver-ApoJ knockout mice will restore "ApoJ sensitivity" is pointless, and invites a reviewer to request that specific experiment be done. I am not, as I am not convinced it will address the underlying principle.

Response: We thank the reviewer for this critical point. As the reviewer pointed out, the experimental evidence for the concept of ApoJ resistance vs ApoJ sensitivity in the current study is limited. Thus, we removed this concept from the manuscript and revised our discussion of the context of ApoJ and LRP2 action.

5) Treatment of PCOS patients with a glitazone could indirectly affect APoJ by effects on hepatic lipoprotein metabolism.

Response: At the reviewer's suggestion, we measured serum LDL-cholesterol, HDL-cholesterol, TG and FFA levels in normal cycling and PCOS subjects before and after pioglitazone treatment. We found that pioglitazone had no significant effect on any of these parameters in PCOS subjects, even as ApoJ levels were reduced. Moreover, we found that circulating ApoJ positively correlated with baseline serum FFA in normal cycling and PCOS subjects. These data are now presented in Table 1 and Fig. 8e. The results are described on page 11, in the 1st paragraph.

Reviewer #2 (Remarks to the Author):

In this manuscript entitled “ApoJ is a novel hepatokine regulating muscle glucose metabolism and insulin sensitivity,” the authors investigated the role of apolipoprotein J (ApoJ) on insulin resistance and glucose intolerance in a low-density lipoprotein receptor-related protein-2 (LRP2)-dependent manner. The authors stated the importance of the ApoJ and LRP2-insulin (IR) axis. This study includes interesting data; however, there are numerous inadequate description and over-interpretation, particularly the relation to LRP2 is suspicious. In this manuscript, the authors showed neither the protein levels of LRP2 nor the direct bound between ApoJ and LRP2. It is also unclear that LRP2 is expressed in skeletal muscle. The authors should show the role of LRP2 in vitro experiments using C2C12 myocytes. Previous reports showed that the expression levels of LRP2 in skeletal muscle are quite low. Therefore this reviewer felt the possibility that the statements include over-interpretation. Other comments are as follows:

Response: We thank the reviewer for these important points. We have revised over-interpretation of LRP2 action throughout the manuscript.

To determine the direct interaction between ApoJ and LRP2, we performed Proximity Ligation Assay (PLA) in C₂C₁₂ cells, a powerful technology of detecting protein interactions with high specificity and sensitivity (Nat Methods 2006; 3: 83-89). We have successfully used this technology to determine the physical interaction of two molecules (Nat Neuroscience 2012; 15: 1392-1398, Nat Communications 2013; 4:1862). As shown in Supplementary Fig. 4g, we found that ApoJ interacts with LRP2 while insulin does not interact with LRP2, indicating a specificity of ApoJ binding to LRP2. Our previous work also demonstrated that ApoJ binds to LRP2 and inhibition of LRP2 expression decreases ApoJ binding to LRP2 in neuronal cells (Nat Communications 2013; 4:1862).

To determine the role of LRP2 in glucose metabolism, we measured insulin-stimulated glucose uptake and Akt phosphorylation in C₂C₁₂ myocytes. Consistent with our in vivo results of muscle-specific LRP2-deficient mice, we found that inhibition of LRP2 in C₂C₁₂ myocytes significantly decreases insulin-stimulated glucose uptake and Akt phosphorylation, suggesting that LRP2 is necessary for the regulation of insulin-mediated glucose metabolism in muscle cells. These data are now presented in Fig. 7d. The results are described on page 10, in the 2nd paragraph.

We further found that insulin increased insulin-stimulated IR internalization in control cells but this effect was impaired when LRP2 expression is suppressed. These data suggest that LRP2 action is required for insulin-induced IR internalization. These data are now presented in Fig. 7b. The results are described on page 10, in the 2nd paragraph.

Major comments:

1. The abbreviation “ApoJ” is not suitable in the title. In addition, I could not find this description in the abstract.

Response: As the reviewer suggested, we have now provided the full name of ApoJ in the abstract and the title.

2. In the introduction, the description of other hepatokine is lacked. Selenoprotein P is known as a hepatokine, and one of its receptor is LRP2. The authors should mention about this hepatokine. I'm wondering that the effects of ApoJ are dependent or independent of selenoprotein P.

Response: As the reviewer suggested, we have now added the description of Selenoprotein P, which is known as a hepatokine, in the introduction. In addition, we measured serum and hepatic Selenoprotein P levels in

control and L-ApoJ^{-/-} mice. We found that hepatic deletion of ApoJ has no effect on Selenoprotein P levels in both serum and liver. These data are now presented in Supplementary Fig. 2f-g. The results are described on page 6, in the 1st paragraph.

3. In the results section, the authors stated “circulating ApoJ is retained in muscle via LRP2 and that LRP2 signaling is essential for maintenance of ApoJ homeostasis in circulation.” It is unclear whether KO of LRP2 is specific in skeletal muscle. The authors should demonstrate this in the LRP2 protein levels. In addition, the authors should show the co-localization of ApoJ with LRP2 by using immunohistochemical method. It is also important to prove this statement by using in vitro experiments.

Response: We thank the reviewer for these important points. We have revised the statement of LRP2 results the reviewer indicated as “circulating ApoJ is retained in muscle via LRP2 and LRP2 signaling could play a role in the maintenance of ApoJ homeostasis in circulation, at least in part.” (page 8, 2nd paragraph)

LRP2 is expressed at a very low levels in muscle. To detect LRP2 in muscle by Western analysis, we isolated the membrane fraction from skeletal muscle. We failed to detect muscle LRP2 protein although we were able to robustly detect LRP2 protein in the kidney. In general, it is technically challenging to detect LRP2 protein by SDS-PAGE due to the huge size of the protein (600kDa). Thus, this approach only works for tissues with abundant receptor expression, such as the kidney, but is typically not adequate for tissues with lower receptor levels.

However, we performed Proximity Ligation Assay (PLA) in muscle, a powerful technology for detecting proteins with high specificity and sensitivity (Nat Methods 2006; 3: 83-89). This technology can detect one single protein with high sensitivity when only one primary antibody is used. Using this technique, we were able to detect LRP2 protein in muscle of control mice but not in muscle of muscle-specific LRP2-deficient mice, although its levels are low, as shown in Supplementary Fig. 4e. To further substantiate, LRP2 expression in muscle, we provided data on LRP2 mRNA levels, documenting that muscle LRP2 mRNA was significantly decreased by ~85% in muscle-specific LRP2-deficient mice compared with control mice in Supplementary Fig. 4b. LRP2 mRNA in other metabolic organs was normal. The results of these data are described on page 7, in the 2nd paragraph.

As the reviewer’s suggestion, we measured the co-localization of ApoJ with LRP2 in muscle by the PLA technology in vivo and in vitro. We found that endogenous ApoJ interacts with LRP2 in muscle of control mice but not in muscle of muscle-specific LRP2-deficient mice. In addition, treatment of C₂C₁₂ cells with ApoJ leads to a significant increase in ApoJ-LRP2 interaction. These data are now presented in Supplementary Fig. 4f-g. The results are described on page 7, in the 2nd paragraph. These results are further supported by our previous work showing that ApoJ binds to LRP2 and inhibition of LRP2 expression decreases ApoJ binding to LRP2 in neuronal cells (Nat Communications 2013; 4:1862).

4. Western blot analysis of ApoJ is plausible because the molecular weight of several blots is not constant.

Response: A notable feature of secreted ApoJ is its heavy glycosylation. It is reported that tissue-specific glycosylation is observed within the same protein (Mol Cell Proteomics. 2015; 14:2103-2110), which could affect the molecular weight of the protein in different tissues when assessing by Western blotting. Fully glycosylated ApoJ in mammalian cells often produces inappropriately disulfide-bonded high molecular weight aggregates (Biochemistry 2007; 46: 1412-1422), which also influence the molecular size of ApoJ protein on the SDS-PAGE. Because of these factors, we think that the sizes of the ApoJ protein in multiple organs could be different when observed on SDS-PAGE/western blotting.

Reviewers' Comments:

Reviewer #1:

Remarks to the Author:

This is a resubmission of a manuscript that describes experimental results supporting a role for liver-secreted ApoJ in maintaining glucose homeostasis. As the authors correctly state, there is a lot of interest in how factors secreted from various cell types participate in homeostasis. In this paper, the authors present evidence that ApoJ secreted by the liver is required for efficient control of glucose levels.

As a resubmission, the authors have made substantial efforts to respond to the reviewers comments. The western blots showing plasma ApoJ levels in mice are in general convincing, and the paper is stronger. However, I still have a few comments:

* To my eye, the phenotype of ApoJ knockout mice indicates a beta-cell problem. The mice have impaired fasting glucose (although the data shown in Fig. 4B is marginal). My reading of Fig 4M is that it confirms higher blood glucose levels, but the impact of insulin is similar (in other words, the slope of the decline). The mice have impaired glucose tolerance (4N), and unless I am mistaken Fig. 4O indicates a failure of glucose-stimulated insulin secretion (at 15min). Part of the problem here may be that ITT is simply not sensitive enough as a tool to measure insulin sensitivity. There is also considerable variability in the signaling data shown in 4P (see for example the pIRS1 data, with 2 WT showing a strong response and 2 showing a very weak response). The authors do report a significant reduction in 2DG uptake in Fig. 6, which shows a modest effect in muscle, adipose and BAT of L-ApoJ^{-/-} mice and a more selective impairment in muscle of M-Lrp2^{-/-} animals. The phenotype shown in Fig. 6A does not however gel with the ITT data. I would suggest removing the ITT data altogether, or doing a clamp study.

* Sure the knockdown of LRP2 in muscle reduces mRNA expression (suppl. fig. 4B - was it significant?). However, LRP2 expression in kidney is >200-fold higher. This suggests to me that the kidney is a primary site of expression, with low/marginal expression in muscle. This makes it hard for me to believe that the ApoJ/LRP2 interaction in muscle is a major driver of ApoJ clearance. The dependence on western blots to measure serum ApoJ is also a weakness - this is a blunt instrument, and is in my opinion not that reliable when trying to measure 10-20% changes. A sensitive EIA is needed.

* The phenotype of Liver ApoJ KO and M-LRP2^{-/-} mice is different. M-LRP2^{-/-} have fasting hyperinsulinemia but normal glucose (in other words, insulin resistance). They have evidence of blunted insulin response (5G), impaired glucose tolerance (5H) but normal insulin levels during the GSIS. As mentioned, to me the phenotype of ApoJ knockout mice indicates a beta-cell problem. I can buy that both phenotypes indicate dysregulation of glucose homeostasis and that this is potentially a quite important finding. However, the phenotypes are to me very different. This should be addressed in the Discussion.

* I don't know what to make of the PCOS data. It is important to support mouse data with clinical observations that are consistent. In this case, there is a nice correlation between ApoJ levels and indices of insulin sensitivity. This is however dependent on using a blunt instrument to measure ApoJ levels - I am assuming this was western blotting as I could not find the method used in the methods section on pg. 12. The decline in ApoJ with TZD treatment is also interesting, but difficult to interpret. This data is, however, important as it support the model. On the other hand, there are commercially available EIA for ApoJ from Thermofisher, RayBiotech, Novus Biologicals and probably others that the authors could have used (and perhaps validated using there genetic models).

Reviewer #2:

Remarks to the Author:

I was fully satisfied with the additional data and improvements. Thus, I would like to recommend the publication of this manuscript.

Response to the Reviewers

Reviewer #1 (Remarks to the Author):

This is a resubmission of a manuscript that describes experimental results supporting a role for liver-secreted ApoJ in maintaining glucose homeostasis. As the authors correctly state, there is a lot of interest in how factors secreted from various cell types participate in homeostasis. In this paper, the authors present evidence that ApoJ secreted by the liver is required for efficient control of glucose levels. As a resubmission, the authors have made substantial efforts to respond to the reviewers comments. The western blots showing plasma ApoJ levels in mice are in general convincing, and the paper is stronger. However, I still have a few comments:

* To my eye, the phenotype of ApoJ knockout mice indicates a beta-cell problem. The mice have impaired fasting glucose (although the data shown in Fig. 4B is marginal). My reading of Fig 4M is that it confirms higher blood glucose levels, but the impact of insulin is similar (in other words, the slope of the decline). The mice have impaired glucose tolerance (4N), and unless I am mistaken Fig. 4O indicates a failure of glucose-stimulated insulin secretion (at 15min). Part of the problem here may be that ITT is simply not sensitive enough as a tool to measure insulin sensitivity. There is also considerable variability in the signaling data shown in 4P (see for example the pIRS1 data, with 2 WT showing a strong response and 2 showing a very weak response). The authors do report a significant reduction in 2DG uptake in Fig. 6, which shows a modest effect in muscle, adipose and BAT of L-ApoJ^{-/-} mice and a more selective impairment in muscle of M-Lrp2^{-/-} animals. The phenotype shown in Fig. 6A does not however gel with the ITT data. I would suggest removing the ITT data altogether, or doing a clamp study.

Response: We thank the reviewer for these important points. As the reviewer pointed out, the slopes of the ITT curves are similar between control and L-ApoJ^{-/-} mice, suggesting possible "normal" insulin sensitivity. However, L-ApoJ^{-/-} mice still have reduced insulin responsiveness (insulin resistance can consist of either or both components) as maximal dose of insulin (2-10 U ip injection) impairs insulin-stimulated glucose uptake and insulin signaling, respectively. We addressed this important point on page 6, in the 2nd paragraph.

As the reviewer suggested, there seems to be something going on with ApoJ and the pancreatic β -cell in L-ApoJ^{-/-} mice that isn't perturbed in the M-LRP2^{-/-} mice, where circulating ApoJ levels are slightly increased and ApoJ signaling is impaired only in skeletal muscle. This issue also discussed on page 14, in the 1st paragraph.

** Sure the knockdown of LRP2 in muscle reduces mRNA expression (suppl. fig. 4B - was it significant?). However, LRP2 expression in kidney is >200-fold higher. This suggests to me that the kidney is a primary site of expression, with low/marginal expression in muscle. This makes it hard for me to believe that the ApoJ/LRP2 interaction in muscle is a major driver of ApoJ clearance. The dependence on western blots to measure serum ApoJ is also a weakness - this is a blunt instrument, and is in my opinion not that reliable when trying to measure 10-20% changes. A sensitive EIA is needed.*

Response: We thank the reviewer for this important point. We have not performed statistical analysis for mRNA levels of LRP2 in muscle of M-LRP2^{-/-} mice because of a small number of animals. However,

we have now added muscle samples and performed statistical analysis for LRP2 mRNA showing a significant difference between control and M-LRP2^{-/-} mice (p < 0.004).

As the reviewer's suggested, we have measured serum ApoJ levels using ELISA, which is more sensitive than immunoblotting analysis. We found that circulating ApoJ levels in M-LRP2^{-/-} mice is increased by ~18% compared with control mice. This new data replaced the old data of Western blotting (~65% increase), which is now shown in Fig. 5a. The results are described on page 8, in the 2nd paragraph. The results are described on page 6, in the 1st paragraph. We appreciate the reviewer's critical point.

** The phenotype of Liver ApoJ KO and M-LRP2^{-/-} mice is different. M-LRP2^{-/-} have fasting hyperinsulinemia but normal glucose (in other words, insulin resistance). They have evidence of blunted insulin response (5G), impaired glucose tolerance (5H) but normal insulin levels during the GSIS. As mentioned, to me the phenotype of ApoJ knockout mice indicates a beta-cell problem. I can buy that both phenotypes indicate dysregulation of glucose homeostasis and that this is potentially a quite important finding. However, the phenotypes are to me very different. This should be addressed in the Discussion.*

Response: We agreed with the reviewer's points. According to the reviewer's suggestion, we have now discussed the difference of metabolic phenotypes between L-ApoJ^{-/-} and M-LRP2^{-/-} mice on page 14, in the 1st paragraph.

** I don't know what to make of the PCOS data. It is important to support mouse data with clinical observations that are consistent. In this case, there is a nice correlation between ApoJ levels and indices of insulin sensitivity. This is however dependent on using a blunt instrument to measure ApoJ levels - I am assuming this was western blotting as I could not find the method used in the methods section on pg. 12. The decline in ApoJ with TZD treatment is also interesting, but difficult to interpret. This data is, however, important as it support the model. On the other hand, there are commercially available EIA for ApoJ from Thermofisher, RayBiotech, Novus Biologicals and probably others that the authors could have used (and perhaps validated using there genetic models).*

Response: We are sorry for missing the information of ApoJ measurements in the method section. We have measured serum ApoJ levels of PCOS sample by immunoblotting analysis and ELISA. ApoJ data measured by ELISA have used for all analyses for correlations. We are now added the information of ApoJ measurement on page 12, in the first paragraph.

Reviewer #2 (Remarks to the Author):

I was fully satisfied with the additional data and improvements. Thus, I would like to recommend the publication of this manuscript.

Response: Thank you very much.

Reviewers' Comments:

Reviewer #1:

Remarks to the Author:

The authors have addressed my concerns, I have no further comments